# Towards Diverse Scientific Hypothesis Search with Large Language Models

**Haorui Wang** [1] [*] **Parshin Shojaee** [2] [*] **Kazem Meidani** [3] [*] **Kunyang Sun** [4] **José Miguel Hernández-Lobato** [5]
**Teresa Head-Gordon** [4] **Jiajun He** [5] [†] **Chandan K. Reddy** [2] [†] **Chao Zhang** [1] [†] **Yuanqi Du** [6] [†]

## Abstract

Large language models (LLMs) are on the rise for accelerating scientific discovery, most recently in advanced tasks such as generating valid scientific hypotheses. Yet in many discovery settings, the goal is not to identify a single best hypothesis since validation can be noisy and expensive, and scientists benefit from a set of high-quality alternative hypotheses that hedge against downstream uncertainty for the best solutions. Nevertheless, commonly used evolutionary search recipes tend to prioritize optimization over exploration in hypothesis generation, and the resulting selection pressure during the search process leads to diversity collapse. Motivated by these limitations, we formulate hypothesis search as a sampling problem, where the objective is to efficiently produce diverse, high-quality hypotheses under a fixed validation budget. Building on this perspective, we propose EvoDiverse, an evolutionary framework inspired by the classical parallel tempering algorithm that searches hypotheses at multiple temperature levels and enables principled information exchange across temperatures to improve exploration without disrupting convergence. Across domains including molecular discovery, equation discovery, and algorithm discovery, our approach consistently improves both hypothesis quality and diversity under the same validation budget, and produces candidates that remain robust under more expensive downstream computational validations.

---

[*]Equal contribution ,[†]Equal advising [1]Georgia Institute of Technology [2]Virginia Tech [3]Carnegie Mellon University [4]University of California, Berkeley [5]University of Cambridge [6]Microsoft Research New England. Correspondence to: Haorui Wang <hwang984@gatech.edu>, Parshin Shojaee <parshinshojaee@vt.edu>, Kazem Meidani <mk.meydani@gmail.com>.

*Proceedings of the 43rd International Conference on Machine Learning*, Seoul, South Korea. PMLR 306, 2026. Copyright 2026 by the author(s).

## 1. Introduction

Large language models (LLMs), auto-regressive generative models pre-trained on massive corpora, have emerged as a dominant paradigm for general-purpose agents due to strong performance across tasks ranging from mathematical reasoning to coding (Chen, 2021; Romera-Paredes et al., 2024; Novikov et al., 2025). Recently, there has been growing interest in using LLMs to accelerate scientific discovery: tool-using agents assisting scientific tasks (Zheng et al., 2023; M. Bran et al., 2024; Huang et al., 2025), "AI scientist" aiming to automate scientific workflows (Boiko et al., 2023; Lu et al., 2024; Mitchener et al., 2025), and frameworks iteratively searching and refining scientific hypotheses (Wang et al., 2025b; Shojaee et al., 2025a; Wang et al., 2025a; Lu et al., 2025; Gan et al., 2025; Du et al., 2025).

However, in scientific discovery, the objective is rarely to identify a single "best" hypothesis (Kuhn, 1962). Instead, the discovery process is fundamentally stochastic and under-determined: simulation results are approximate, lab experiments are expensive, and multiple competitive hypotheses can remain plausible across the hierarchical validation procedure (Chamberlin, 1890; Renz et al., 2019). As a result, a key desideratum for hypothesis search is not only quality, but also diversity—producing a set of high-quality yet meaningfully distinct hypotheses that hedge against uncertainty in future validation.

Despite impressive empirical progress, most existing LLM-based hypothesis search pipelines implicitly prioritize optimization over exploration and underemphasize diversity. A common recipe is to use LLMs as evolutionary operators in an evolutionary algorithm (EA) (Rechenberg, 1989; Holland, 1992), iteratively proposing better hypotheses, scoring them using the objective function, and selecting top-ranked hypotheses until convergence or running out of evaluation budget. Although simple and effective, this practice induces strong selection pressure that concentrates probability mass in a narrow region of the hypothesis space, leading to limited exploration, premature convergence, and low sample diversity (Lehman & Stanley, 2011). In settings where downstream experiments introduce substantial uncertainty and where multiple distinct hypotheses may be valuable, this collapse of diversity becomes a critical bottleneck.

An alternative perspective is to consider the search for scientific hypotheses as a sampling problem, where the goal is to broadly sample competing hypotheses from a diverse population with probability corresponding to their quality. However, sampling from this distribution with LLMs is highly challenging and often not the right abstraction for a scientific hypothesis search for several reasons: (1) exact sampling with an LLM's proposal requires access to normalized probabilities over the space of possible hypotheses, which is combinatorially large and cannot be meaningfully normalized; (2) even under a constrained hypothesis space, token-level likelihoods are only accessible for open-source models; and (3) the limited validation budget typically prevents asymptotic convergence of common sampling algorithms. More fundamentally, exact sampling is rarely necessary in scientific discovery, since evaluation itself is uncertain, approximate, and often stochastic, meaning that the target distribution is not precisely specified. This motivates a different view: *the objective is neither pure optimization, which collapses diversity, nor exact sampling, which is computationally infeasible, but rather the efficient generation of diverse, high-quality hypotheses under limited validation budgets.*

In this paper, we take an important step towards this goal. We retain the sampling perspective to formulate a scientific hypothesis search, but instead of enforcing exact sampling, we perform an evolutionary search to be approximated by sampling from a Boltzmann distribution with an evolving power factor. Based on this perspective, we propose EvoDiverse, a parallel-tempered evolutionary framework inspired by the classical parallel tempering (PT) sampling algorithm. By running evolutionary searches at two or more temperature levels and enabling principled information exchange across temperatures, higher-temperature populations encourage exploration of the hypothesis space, while lower-temperature populations apply stronger selection to refine promising hypotheses generated by the cross-temperature exchange. Under the same validation budget, we observe that the standard LLM-based evolutionary search exhibits systematic diversity collapse across a wide range of scientific discovery problems. In contrast, EvoDiverse consistently improves both hypothesis quality and diversity, thus producing promising candidate sets for molecular discovery beyond a known chemical space while maintaining drug-like properties and activities, equation discovery evaluated on the LLM-SRBENCH (Shojaee et al., 2025b) benchmark, and algorithm discovery for constrained optimization.

## 2. Background

### 2.1. Evolutionary Algorithm

We consider a (single-objective) optimization problem:

$$x^* \in \arg\min_{x \in \mathcal{C}} h(x), \tag{1}$$

where $h : \mathcal{C} \to \mathbb{R}$ is the objective function and $\mathcal{C}$ is the optimization domain. EAs are population-based, derivative-free methods for black-box optimization that mimic the natural evolution process (Rechenberg, 1989; Holland, 1992). EAs maintain a population of candidates which evolve in each generation (iteration) utilizing a general-purpose algorithm: (1) create mating pool: sample parents from current population; (2) mate: produce a set of offspring by applying mutation and/or crossover operators to the parents; (3) score: validate candidates using the objective function; (4) select: prepare new population based on scores and preferences over all candidates.

**LLM-augmented EAs**. In LLM-augmented EA frameworks, LLMs instantiate the mating operator: given a set of parent solutions, it proposes new offspring, while the objective function provides the scores used for selection.

### 2.2. Parallel Tempering (PT)

The goal of a sampling task is to draw samples from an (often unnormalized) probability density function. One famous probability function from the physical sciences is the Boltzmann density:

$$p(x) \propto \exp(-\beta U(x)) \tag{2}$$

where $U(\cdot) : \mathcal{C} \to \mathbb{R}$ is the potential energy function, and $\beta$ is the inverse temperature parameter. Note that algorithms based on Boltzmann sampling can be used to solve optimization problems in the limit $\beta \to \infty$ by replacing the energy function with an arbitrary objective function (Hwang, 1980; Gelfand & Mitter, 1991; Raginsky et al., 2017; Ma et al., 2019).

PT is a sampling algorithm (Swendsen & Wang, 1986; Geyer, 1991; Hukushima & Nemoto, 1996) known to be efficient in sampling multimodal distributions.(Earl & Deem, 2005) Instead of running Markov chain Monte Carlo (MCMC) algorithms to mix the target distribution directly, PT defines a variable temperature sequence of Boltzmann distributions and mixes the joint (product) distribution of all temperature levels. Hence PT performs local exploration at each temperature using MCMC, while global exploration is encouraged through Boltzmann distributions at higher temperatures (small $\beta$) that communicate their diverse samples via swaps with samples generated at lower temperatures (latge $\beta$). To ensure the sampled distributions are correct, the Metropolis-Hastings algorithm is also applied to the communication step: concretely, consider the target distribution at two levels $p_1(x) \propto \exp(-\beta_1 U(x))$ and $p_2(x) \propto \exp(-\beta_2 U(x))$. The joint states before and after the swap are defined as $X = (x_1, x_2)$ and $X' = (x_2, x_1)$, and the acceptance ratio for the swapping step is defined as

$$A(X, X') = \min \left\{ 1, \frac{\exp(-\beta_1 U(x_2) - \beta_2 U(x_1))}{\exp(-\beta_1 U(x_1) - \beta_2 U(x_2))} \right\} \tag{3}$$

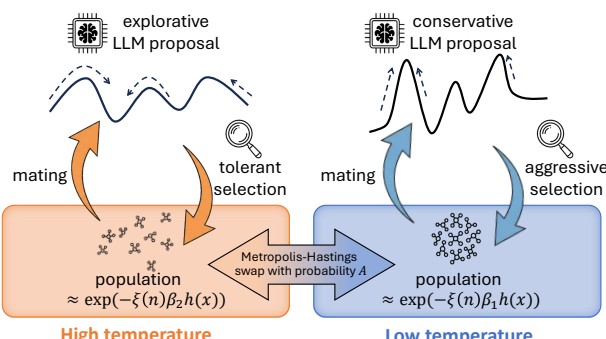

*Figure 1.* Illustration of our parallel tempered evolutionary algorithm. High-temperature EA explores the hypothesis space faster, while low-temperature EA converges to local minima faster. The swap between the two pools encourages exploration at the lower temperature. Swapping with Metropolis–Hastings provides a cleaner communication mechanism than direct migration: it exchanges individuals only when the move is approximately consistent with both temperatures, so information can flow across temperatures without disrupting convergence of the EA process.

This ensures that the stationary distributions on the joint states are defined as $p(X) = p(x_1, x_2) := p_1(x_1)p_2(x_2)$. Intuitively, the communication step redistributes samples to appropriate intermediate distributions that obey Boltzmann weighted states.

# 3. Methodology

A key limitation of standard evolutionary algorithms is that, although they are intended for global optimization, they can readily become trapped in local optima, leading to premature convergence and poor exploration of diverse solutions. Inspired by PT sampling, we introduce a mechanism to enhance diversity of EA by maintaining multiple populations at different values of the temperature hyperparameter and frequently exchanging candidate samples across the populations held at different temperatures. We design the exchange rule so that it largely preserves the convergence behavior of the EA. The remainder of this section proceeds as follows. We first explain how we approximate the distribution induced by the evolving population, and then present the temperature ladder and swap operator, followed by the complete algorithm and practical implementation details.

## 3.1. Evolutionary Search as Approximate Sampling

EA is a global optimization algorithm that seeks the global optima by iteratively mating, mutating, and selecting with stochasticity. We can view this iterative process as an approximate sampling algorithm, and the population at each iteration follows approximately the following Boltzmann distribution:

$$p(x) \propto \exp(-\xi(n)\beta h(x)) \qquad (4)$$

where $h$ is the objective function to minimize, $\beta$ is a factor reflecting the selection strength and $\xi(n)$ is a factor that generally increases with iteration $n$. An intuitive way to understand this dynamic $\xi(n)\beta$ is to consider a trivial EA, where we start with a uniform distribution over a large population and only conduct selection by sampling according to the score, scaled by $\beta$ at each iteration. In this case, the distribution of population samples follows simply as $\exp(-n\beta h(x))$.

While the trend of $\xi(n)$ is increasing, the rate at which it increases can vary widely, depending heavily on the EA's convergence speed. The rate at which it is increasing also has a large effect on the final performance: increasing faster will generally encourage convergence but lack exploration, while slower ones will present better exploration and coverage, but might fail to converge within a limited budget.

## 3.2. Parallel Tempered Evolutionary Algorithm

*Can we both achieve good convergence and sample diversity?* Inspired by PT, we define a ladder of temperature annealed distributions to help exploration in order to to improve EA. In particular, we create a tempered distribution at higher temperatures and use a more tolerant selection to encourage more aggressive exploration; at lower temperatures, we apply harsher selection and conservative exploration. We then design a communication mechanism between the temperature rungs so that promising hypothesis samples discovered at higher temperatures can propagate to, and improve, the search at lower temperatures. For simplicity, we consider the scenario with only two temperatures, but our method generalizes to efficient reversible PT schemes(Brown & Head-Gordon, 2003), multiple temperature levels, and non-reversible PT also could apply(Syed et al., 2022).

### 3.2.1. TEMPERED EVOLUTIONARY SEARCH

We first consider the EA search at each temperature. In Wang et al. (2025b), the selection in EA is implemented in a deterministic way: we select the top-$\nu$ samples from the current population to form a mating pool, produce $o$ offsprings from parents from the current population, and select the top-$N$ candidates across the previous population and the new offspring for the next population.

However, because the selections are fully deterministic, it is difficult to tune selection pressure across temperatures, i.e., to make selection more aggressive at low temperatures and more tolerant at high temperatures. Therefore, noting the core steps of EAs introduced in Section 2.1 can be implemented flexibly, we adopt a stochastic rule that samples

candidate $x_i$ *without replacement* with probability

$$p(x_i) = \frac{\exp\big(-h(x_i)\big)^{\beta}}{\sum_{k=1}^{K} \exp\big(-h(x_k)\big)^{\beta}}, \quad (5)$$

where $\beta$ is controlling the temperature, which directly influences the selection strength. In particular, higher temperatures correspond to a smaller $\beta$ to encourage exploration, while lower temperatures correspond to a larger $\beta$. Note that when $\beta \to \infty$, this recovers the deterministic choice, while when $\beta \to 0$, there is no selection pressure.

In addition, we may also use different prompts for the LLM at different temperatures. For example, in chemical discovery tasks, the high-temperature prompt we use encourages proposing structurally diverse molecules and novel scaffolds while maintaining competitive target scores, whereas the low-temperature prompt steers the model toward refining known high-performing motifs and proposing candidates with improved predicted activity.

### 3.2.2. SWAP OPERATOR

We now consider how to communicate between the higher and lower temperatures. A straightforward approach is to directly exchange samples between adjacent temperatures: we draw a subset of candidates from the lower- and higher-temperature populations and insert them into each other's pools. This is similar to the Island strategy in EA, typically employed by running multiple replicas to solve the same optimization problem. However, when pools operate at different temperatures, naively exchanging samples can be problematic. Most high-temperature samples may have lower scores than low-temperature ones, so injecting them into the low-temperature pool may result in their immediate discard. Conversely, low-temperature samples typically score better; if moved into the high-temperature pool, they can dominate the population and interrupt exploration.

Therefore, we need to design a swapping mechanism where the distribution of different temperatures can be largely maintained. Inspired by PT, we apply the Metropolis-Hastings algorithm over the approximated distribution for each temperature. In each iteration, the pools in each level are *approximately* sampled from the following distributions:

$$\text{cold pool:} \quad p_1(x) \propto \exp(-\xi(n)\beta_1 h(x)), \quad (6)$$
$$\text{hot pool:} \quad p_2(x) \propto \exp(-\xi(n)\beta_2 h(x)) \quad (7)$$

where $\beta_2 < \beta_1$, reflecting that the higher-temperature population is more tolerant, resulting in a flatter distribution, and $\xi(n)$ reflects the distribution of both pools changing along the iteration $n$. The joint state formed by the two pools is approximately distributed as

$$(x_1, x_2) \sim p_1(x_1)p_2(x_2). \quad (8)$$

---

**Algorithm 1:** Parallel tempered evolutionary algorithm (detailed version in Appendix Algorithm 2)

**Input:** Initial populations at low and high temperatures
**Output:** Final populations

1 Initialize populations at two temperatures;
2 **for** *each iteration* **do**
3     // Evolve within each temperature
    **for** *each temperature level* **do**
4        Select parents;
5        Generate offspring using LLM operator;
6        Evaluate and update population;
    // Exchange information across temperatures
7     **if** *swap step* **then**
8        Propose swaps;
9        Accept swaps using Metropolis-Hastings rule;
10        Adapt parameter $\xi$ for a stable swap rate;
11 **return** final populations;

---

We seek a swapping operator that preserves this product target: if $(x_1, x_2) \sim p_1(x_1)p_2(x_2)$ before swapping, then the joint distribution remains stationary after applying the operator. This can be ensured by imposing detailed balance with respect to $p_1(x_1)p_2(x_2)$. In PT, this is achieved by a Metropolis-Hastings (MH) move:

- **Propose a swap:** the proposed new state reuses the current samples and simply swaps their temperature assignments: $(x_1', x_2') \leftarrow (x_2, x_1)$.

- **Compute the acceptance ratio:**

$$a = \frac{p_1(x_1')\, p_2(x_2')}{p_1(x_1)\, p_2(x_2)} = \frac{p_1(x_2)\, p_2(x_1)}{p_1(x_1)\, p_2(x_2)} \quad (9)$$
$$= \exp(-\xi(n)(\beta_1 - \beta_2)(h(x_2) - h(x_1))) \quad (10)$$

- **Accept/reject:** accept the swap with probability $A = \min\{1, a\}$, or otherwise keep $(x_1, x_2)$ unchanged.

**Aligning convergence of different temperature levels**. We have outlined the overall framework so far. However, computing the swap acceptance probability requires access to $\xi(n)$, which is typically unknown and intractable, as it may depend on the LLM, the prompt, and other implementation details. This is also where our setting departs from classical PT: rather than targeting fixed stationary distributions with known density functions, each temperature level corresponds to some unknown distribution that is continuously sharpened over time, potentially at different rates.

To choose $\xi(n)$ and align the convergence rates across temperatures, we adopt the following heuristics. Our goal is to

ensure that the swapping mechanism remains effective even as both temperature levels evolve and converge, possibly at mismatched speeds. Accordingly, we treat $\xi$ as a dynamic hyperparameter that regulates the swap rate and keeps it within a roughly constant, well-behaved range (e.g., 30%-50%). We hence tune this parameter on the fly by keeping track of the swap rate across recent iterations and modify the value of $\xi$ accordingly.

**Practical benefit of swapping**. Intuitively, PT moves candidates toward the temperature level that best matches their quality. The PT swap operator does not discard candidates; instead, it reallocates them across temperatures so that strong candidates tend to remain in lower-temperature pools, while weaker or more diverse candidates can be retained at higher temperatures for further refinement. At high temperatures, selection is more tolerant, so the effective pool size is larger. Many candidates have a greater chance of entering the mating pool, thereby promoting broader exploration over a larger region of the search space. PT can also be seen as a more principled way to design communication in island EA (Whitley & Starkweather, 1990; Mühlenbein, 1991).

## 4. Experiment

In this section, we focus on three different discovery problems: molecular discovery, equation discovery across biology, physics, chemistry and materials science, and algorithm discovery. We conduct comprehensive quantitative and qualitative evaluations [1] on the quality and diversity of the generated solutions under the same evaluation budget.

### 4.1. Molecular Discovery

**Task Description.** The JNK3 and GSK3$\beta$ benchmarks formulate hit identification as a *single-objective* optimization task over chemical space. For each target, the objective is to sample molecules that maximize a scalar oracle score representing the predicted inhibitory activity (Huang et al., 2021). Following the standard protocol used in MOLLEO, the optimization starts from an initial population of 120 molecules sampled from ZINC-250K (Irwin & Shoichet, 2005) and is conducted under a fixed budget of 10,000 oracle calls (with the same early-stopping criterion as prior work) (Wang et al., 2025b). We utilize Deepeek V3.2 (Liu et al., 2025) as our LLM operator.

**Comparison methods.** We compare EvoDiverse against three variants of LLM-guided evolutionary search *under the same evaluation budget*: (1) **MOLLEO** (Wang et al., 2025b): a standard evolutionary algorithm with LLMs acting as evolutionary operators (e.g., mutation/crossover) that achieves state-of-the-art performance on the PMO bench-

mark (Gao et al., 2022); (2) **Ensemble**: a MOLLEO variant that maintains two isolated populations throughout optimization, and reports results by merging their final candidate sets to form the overall output; (3) **Tempering**: a single-population hot-temperature variant, where the evolutionary search is identical to MOLLEO except that the LLM evolutionary operators are sampled at a higher temperature to encourage more exploration, (4) **EvoDiverse**: we adapt MOLLEO to two temperature levels (cold and hot) with our parallel tempered evolutionary algorithm framework, detailed in Appendix B.2.2.

**Evaluation metrics.** We report two metrics: *diversity-aware Top-10 AUC* (Top-10 AUC) and *diversity-aware Top-10 average score* (Top-10 Avg). Both the formal definitions of these metrics and the selection rule used to construct the Top-10 candidate set are detailed in Appendix B.2.1.

**Results.** Compared to the MOLLEO baseline that does not have an additional hot pool for sample swapping, EvoDiverse maintains a higher average score among candidates while preserving a high volume of diverse samples for both JNK3 and GSK3$\beta$ tasks (Figure 2(a) and Appendix Figure 5(a)). Moreover, MOLLEO struggles to optimize oracle scores in early iterations, while EvoDiverse is able to quickly narrow down to high scores by sacrificing a minimal portion of diversity. At convergence, EvoDiverse consistently proposes approximately 90 diverse candidates, nearly doubling the $\sim$50 samples produced by MOLLEO (Figure 2(a)). This substantial margin further proves the efficacy of the hot pool in guiding the optimization process toward superior average scores without collapsing into a local chemical space.

In real-world drug discovery, candidates must be both potent and structurally diverse to hedge against downstream uncertainty in efficacy. We demonstrate that EvoDiverse identifies high-quality candidates without "hacking" the reward function, consistently retaining high drug-likeness (QED) and synthesis accessibility (SA) scores (Figure 2(b)) despite not explicitly optimizing them. This result shows that LLM might internally have the knowledge to correlate binding capability with physicochemical feasibility. Beyond maintaining drug-likeness, t-SNE analysis relative to ChEMBL dataset (Mendez et al., 2019) reveals that EvoDiverse traverses known chemistry to explore entirely novel regions of chemical space (Appendix Figure 6). This observation confirms that our diversity-aware selection enables targeted exploration into new, synthesizable chemical territory rather than mere rediscovery of known molecules.

The integration of the hot pool and swapping algorithm allows EvoDiverse to exhibit the fastest convergence across all targets, as quantified by Top-10 AUC (Table 1). While Ensemble baselines provide variety, they lack an effective mechanism for progress, often remaining in low-score regions (Appendix Figure 5(b)). The advantage of this ar-

---

[1]Our code is available at https://github.com/zoom-wang112358/EvoDiverse.

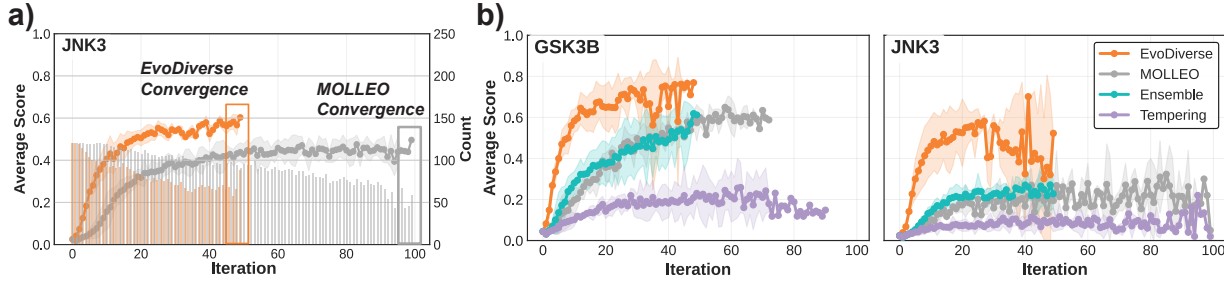

*Figure 2.* Optimization procedure visualization for molecular discovery. (a) Evolution of average JNK3 scores (lines) and the corresponding number of samples (bars) passing diversity-aware selection for MOLLEO and EvoDiverse. (b) Optimization trajectories of average GSK3$\beta$ and JNK3 scores compared against baselines for molecules satisfying QED $> 0.5$ and SA $< 5.5$ from each iteration.

chitecture is further highlighted by our GraphGA-adapted version (Appendix Table 5), which utilizes samples more efficiently and identifies optimal directions significantly faster than the vanilla GraphGA(Jensen, 2019) baseline.

*Table 1.* Quantitative evaluation for molecular discovery tasks.

| Target | Method | Metrics | |
|---|---|---|---|
| | | Top-10 AUC ↑ | Top-10 Avg ↑ |
| JNK3 | MOLLEO | $0.58 \pm 0.05$ | $0.66 \pm 0.06$ |
| | Ensemble | $0.54 \pm 0.04$ | $0.71 \pm 0.03$ |
| | Tempering | $0.46 \pm 0.04$ | $0.59 \pm 0.09$ |
| | EvoDiverse | $\mathbf{0.63 \pm 0.05}$ | $\mathbf{0.74 \pm 0.04}$ |
| GSK3$\beta$ | MOLLEO | $0.70 \pm 0.03$ | $0.82 \pm 0.03$ |
| | Ensemble | $0.73 \pm 0.04$ | $\mathbf{0.85 \pm 0.04}$ |
| | Tempering | $0.58 \pm 0.02$ | $0.70 \pm 0.07$ |
| | EvoDiverse | $\mathbf{0.76 \pm 0.02}$ | $0.82 \pm 0.03$ |

### 4.2. Equation Discovery

**Task Description.** Scientific equation discovery (symbolic regression) aims to recover concise and interpretable mathematical laws from observational data. Given measurements $\{(x_i, y_i)\}_{i=1}^{N}$, where $x_i \in \mathbb{R}^d$ denotes system variables and $y_i \in \mathbb{R}$ the observed response, the objective is to infer a symbolic function $\hat{f}(x)$ that both fits the data and extrapolates beyond the training regime. Unlike black-box regression, equation discovery constitutes a structured, combinatorial search over a discrete hypothesis space of symbolic programs, where successful solutions must jointly satisfy numerical accuracy, interpretability, and semantic validity. In our experiments, we evaluate this task on LLM-SRBENCH (Shojaee et al., 2025b), a recent standard benchmark for LLM-based equation discovery frameworks which spans across various scientific domains (physics, biology, chemistry, and materials science). Across all experiments, we use a fixed evaluation budget of 1,000 program evaluations and instantiate each method with two representative LLM backbones, DeepSeek-V3.2 and GPT-5 for fair comparison.

**Comparison methods.** We compare EvoDiverse on equation discovery against the following representative baselines:

*Table 2.* Quantitative evaluation on equation discovery across LLM-SRBench problems, evaluating both diversity and quality of hypotheses. Results are averaged over all datasets per domain and across DeepSeek-V3.2 and GPT-5 backbones.

| Dataset | Method | Metrics | | |
|---|---|---|---|---|
| | | Diversity ↑ | Best $Acc_{0.1}$↑ | Top-10 $Acc_{0.1}$↑ |
| Physics | Tempering | 0.294 | 0.301 | 0.237 |
| | Ensemble (LLM-SR) | 0.287 | 0.283 | 0.252 |
| | EvoDiverse | **0.305** | **0.408** | **0.275** |
| Biology | Tempering | 0.257 | 0.104 | 0.096 |
| | Ensemble (LLM-SR) | 0.255 | 0.104 | 0.132 |
| | EvoDiverse | **0.290** | **0.212** | **0.146** |
| Chemistry | Tempering | 0.265 | 0.506 | 0.350 |
| | Ensemble (LLM-SR) | 0.265 | 0.588 | 0.386 |
| | EvoDiverse | **0.284** | **0.618** | **0.433** |
| Materials | Tempering | 0.209 | 0.694 | 0.713 |
| | Ensemble (LLM-SR) | 0.215 | 0.699 | 0.757 |
| | EvoDiverse | **0.223** | **0.803** | **0.763** |

(1) **Tempering**: a single-population LLM-guided evolutionary search baseline with high-temperature sampling; (2) **Ensemble** (Shojaee et al., 2025a): a multi-island framework, similar to the well-known LLM-SR (Shojaee et al., 2025a) framework, with multiple independent populations evolving in parallel without interaction; and (3) **EvoDiverse**: our proposed diversity-promoting evolutionary framework with two temperature (cold and hot) pools. All LLM-based methods share the same program grammar, prompt structure, numerical fitting procedure, and evaluation protocol, differing only in how candidate populations are organized and reused.

**Evaluation Metrics.** We evaluate both *quality* and *diversity* of discovered hypotheses in equation discovery. Quality is measured using the *Best Score* (as lowest normalized mean square error (NMSE), and highest accuracy to threshold ($Acc_{0.1}$) (Shojaee et al., 2025a) averaged across LLM backbones and datasets within each category), and the Top-10 score, which reflects the mean performance of the ten best hypotheses at convergence across runs (performance reported as error so the lower the better). To quantify exploration, we measure *program diversity* as the

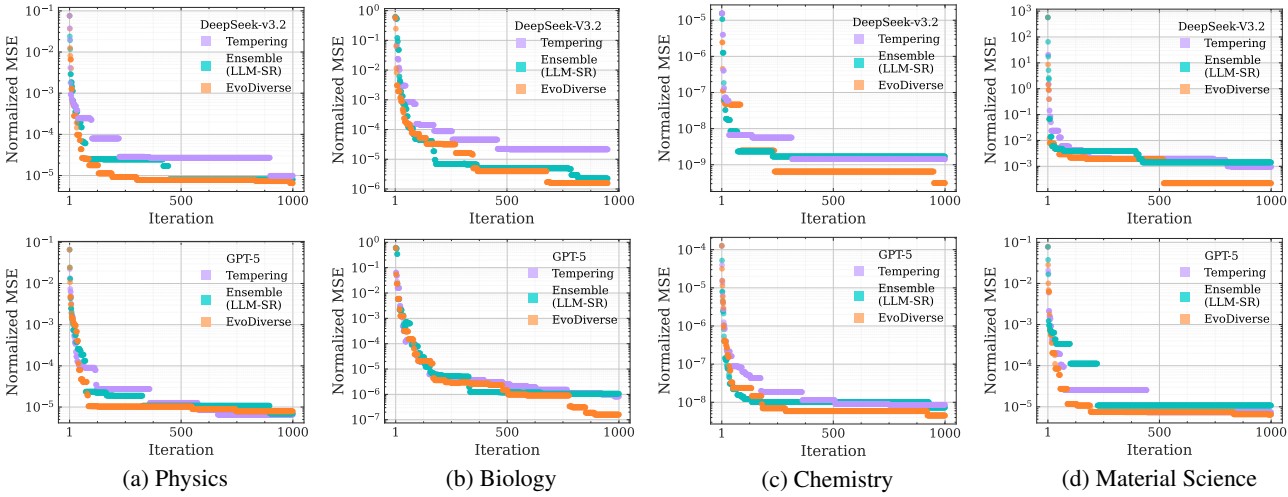

*Figure 3.* Comparison of EvoDiverse and baselines along best-score trajectories in equation discovery (normalized error; lower is better).

mean pairwise cosine distance between CodeBERT (Feng, 2020) embeddings of symbolic programs. Higher values indicate broader coverage and diversity of the explored hypothesis space. Unless otherwise stated, results are averaged across datasets within each category and LLM backbones; detailed per-dataset and per-backbone results for more metrics are provided in the Appendix B.3.

**Results.** Table 2 summarizes the overall performance across all datasets, reporting averages over both LLM backbones to conserve space (full results in Appendix B.3). EvoDiverse consistently achieves substantially higher diversity than all baselines, while maintaining comparable, and often superior, best-score and Top-10 scores. This indicates that the additional diversity introduced by EvoDiverse is not incidental, but *productive* which enables the search to explore more informative regions of the hypothesis space that lead to higher-quality equations. EvoDiverse's advantage is further reflected in optimization dynamics. Figure 3 shows the evolution of the best error over iterations. Across all dataset categories and both LLM backbones, Evo-Diverse exhibits a consistently lower error curve, indicating faster convergence and better final solutions. In contrast, single-pool tempering often suffers from premature collapse, while ensemble methods exhibit slower progress due to lack of information exchange between the populations. For additional results across datasets, model backbones, and the comparison with state-of-the-art non-LLM baseline PySR (Cranmer, 2023), check Appendix B.3.

To better understand these effects, we provide a detailed diversity analysis in the appendix (check Tables 15-16 and Figure 10), including diversity statistics over the full buffer and elite candidates, as well as embedding-space visualizations of discovered programs. Qualitatively, EvoDiverse exhibits broader and more structured coverage of the program embedding space, while baselines tend to cluster tightly or fragment into isolated regions. These

results confirm that the systematic swapping mechanism in EvoDiverse encourages *high-quality diversity*, i.e., diversity that directly supports more effective scientific discovery rather than merely increasing variance.

### 4.3. Algorithm Discovery

**Task Description.** The circle packing optimization problem requires placing $n$ circles within a unit square such that the sum of their radii is maximized while ensuring no circles overlap and all remain fully contained within the boundary. We focus on the $n = 26$ case, a canonical benchmark in algorithm discovery (Romera-Paredes et al., 2024; Liu et al., 2024; Novikov et al., 2025; Sharma, 2025). This constrained optimization challenge combines discrete placement decisions with continuous radius optimization, creating a rugged fitness landscape with multiple local optima that tests the exploration-exploitation trade-off of evolutionary algorithms. We set the budget as 1,000 program evaluations and use DeepSeek-V3.2 as our LLM backbone (additional details in Appendix B.4.1).

**Comparison methods.** We compare EvoDiverse against three baselines: (1) **EA**: a standard evolutionary algorithm with one population; (2) **Island**: two populations with frequent migration, representing heuristics-based interaction; (3) **Ensemble**: two isolated populations, but with no interaction; and (4) **EvoDiverse**: our parallel tempered framework with two temperature levels (cold and hot) and principled Metropolis-Hasting swaps.

**Results.** Table 3 presents the main results. EvoDiverse achieves the best optimization performance across all methods, demonstrating that controlled thermodynamic exchange effectively balances exploration and exploitation. Among the baselines, Island achieves the lowest diversity, indicating population homogenization from frequent migration. Conversely, Ensemble maintains high overall diversity

*Table 3.* Circle packing results ($n = 26$). Diversity measured via TF-IDF embeddings; comprehensive metrics including semantic diversity (via CodeBERT) in Appendix B.4.2.

| Method | Best Sum | Top-100 Avg | Diversity |
|---|---|---|---|
| EA | 2.4986 | 2.4302 | 0.61 |
| Island | 2.4247 | 2.4241 | 0.48 |
| Ensemble | 2.4105 | 2.3330 | 0.76 |
| EvoDiverse | **2.5461** | **2.5138** | **0.78** |

by keeping populations distinct, but delivers weak final performance, suggesting that principled communication between populations is essential. EA falls between these extremes but lacks a principled mechanism to manage the exploration-exploitation trade-off.

Figure 4 shows that EvoDiverse maintains consistent improvement throughout the search horizon, while baselines exhibit earlier plateaus. The performance gap between EvoDiverse and Ensemble and Island baselines demonstrates that the principled communication is more effective than heuristic-based approach which can lead to either premature or late convergence. We also annotate the solution evolution found by EvoDiverse.

Detailed analysis in Appendix B.4.2 reveals further insights into these mechanisms. Per-pool diversity metrics (Appendix Table 23) show that EvoDiverse's Cold and Hot pools maintain distinct characteristics, with the Cold pool producing the majority of elite solutions, evidence of successful funneling from exploration to refinement. In contrast, Island's pools become nearly identical, while Ensemble exhibits severe asymmetry where only one pool contributes to the final elite set. PCA visualization (Appendix Figure 13) qualitatively confirms these patterns.

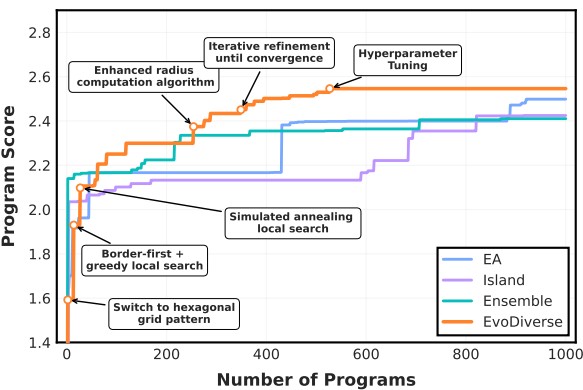

*Figure 4.* Optimization trajectories for circle packing problem.

## 5. Related Work

**LLMs for Scientific Hypothesis Search**   Many works have explored using LLMs to search for new hypotheses to accelerate scientific discovery. A prominent line treats LLMs as heuristics in which an LLM proposes hypothe-

sis candidates while an external oracle scores them, often instantiated as an evolutionary search. FunSearch (Romera-Paredes et al., 2024) is a representative example that focuses on mathematical programs, and similar ideas have been considered for symbolic regression (Grayeli et al., 2024; Shojaee et al., 2025a), molecular design (Wang et al., 2025b), materials discovery (Gan et al., 2025; Lu et al., 2025) and protein optimization (Chen et al., 2024; Wang et al., 2025c), and even has been extended to solving sequential decision making such as retrosynthesis planning (Wang et al., 2025a). Complementary approaches consider LLMs to iteratively analyze and refine their own hypotheses or interact with human experts (Liu et al., 2023; Yuksekgonul et al., 2025). Recent benchmarks have begun to systematically evaluate the abilities of LLMs in searching scientific hypotheses (Song et al., 2025b).

**Test-time Scaling for LLMs**   Beyond evolutionary search used in scientific discovery, numerous approaches have been studied to scale LLMs at test-time. Early work improved downstream performance by eliciting intermediate reasoning traces, including in-context learning and chain-of-thought prompting (Brown et al., 2020; Wei et al., 2022). Another branch of methods directly controls the decoding process and applies selection mechanisms, including self-consistency (Wang et al., 2022), verification (Cobbe et al., 2021), confidence filtering (Fu et al., 2025) and more (Leviathan et al., 2023; Du et al., 2023). Among them, sampling-based approaches directly target test-time distributions, including reward-tilted (Mudgal et al., 2023) and tempered distributions (Karan & Du, 2025). Other alternatives are representation-based approaches that seek to steer LLMs by understanding and intervening through their internal mechanisms (Cunningham et al., 2023; Li et al., 2023; Kong et al., 2024).

**Diversity for Evolutionary Algorithms**   Prior work in evolutionary algorithms has explored improving solution diversity via a range of mechanisms, including quality diversity (QD) methods (Pugh et al., 2016) and island-based EAs (Whitley & Starkweather, 1990; Mühlenbein, 1991). Our method can be viewed as a probabilistic island EA: each island (temperature level) can influence others more effectively through communication. Moreover, our framework is compatible with QD, which can be incorporated into the EA objectives to further encourage exploration.

**Parallel Tempering in Generative Models**   Recently, PT has been adapted to generative models as a mechanism to control and scale the generative process. He et al. (2025) demonstrated that PT-based strategies can yield substantially greater diversity than alternative sequential Monte Carlo methods for inference-time control of diffusion models.

## 6. Conclusion, Limitation and Future Works

In this paper, we advocate the importance of exploring a wide range of candidates in searching for scientific hypotheses. We view LLM-based scientific hypothesis search from a sampling perspective and propose a parallel-tempered evolutionary framework inspired by parallel tempering. Both quantitative and qualitative validations demonstrate the effectiveness of this new approach across multiple scientific discovery scenarios and show that diversity yields advantages when evaluations are unknown.

**Limitations** EvoDiverse introduces a budget-dependent trade-off: multiple temperature pools improve exploration and mitigate diversity collapse, but under a fixed oracle budget they reduce the number of evaluations available to each pool. Therefore, the number of pools, temperature gap, and swap frequency remain practical hyperparameters that may need task-specific tuning.

The method is also an approximate parallel-tempering procedure. The EA population induced by LLM proposals does not follow a known stationary distribution, and the swap rule can be sensitive to the objective scale. For example, equation discovery requires a log-MSE energy transformation for stability (see Section B.3.6). Automating such energy and ladder choices is left to future work.

**Future Works** The practical deployment of the algorithm requires an understanding of the search space and objectives to properly set the Boltzmann constant to convert them into probabilities. Despite promising computational results, the validation of the hypotheses found for future work still requires real-world experiments for further validation. In addition, it is worth exploring the algorithmic design space to improve the sample diversity of EA and other commonly used search algorithms, such as tree search (Song et al., 2025a). The study of how fine-tuning can further improve the sample diversity of LLMs in searching scientific hypotheses is another promising future direction (Cavanagh et al., 2026; Sun et al., 2025).

## Impact Statement

This paper advances machine learning and accelerates scientific discovery and may have societal impacts, none of which we believe require specific discussion.

## Acknowledgement

T.H-G. thanks the CPIMS program, Office of Science, Office of Basic Energy Sciences, Chemical Sciences Division of the U.S. Department of Energy under Contract DE-AC02-05CH11231 for support of AI methods. T.H-G. and K.S. were supported by the National Institute of Allergy and Infectious Disease grant U19-AI171954 for the drug discovery application. JH acknowledges support from the University of Cambridge Harding Distinguished Postgraduate Scholars Programme. JMHL acknowledges funding from AI Hub in Generative Models, under grant EP/Y028805/1.

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

# Appendix:
# Towards Diverse Scientific Hypothesis Search with Large Language Models

## Appendix Contents

## A. Pseudocodes

In this section, we include the detailed algorithm of our approach. We first define the function `SampleWithoutReplacement(set=`$\mathcal{A}$`, size=`$a$`, score=`$h(\cdot)$`, power=`$\beta$`)` as drawing $a$ samples without replacement from set $\mathcal{A}$, where each sample is assigned with probability defined in Equation (5). Then we detail the algorithm in Algorithm 2. Note that Algorithm 2 gives the generic parallel-tempered evolutionary template; in the experiments, the score-to-weight conversion, parent-selection rule, and survivor-update rule are instantiated separately for each domain, as summarized in Appendix B.1.

---

**Algorithm 2:** Parallel tempered evolutionary algorithm

---

**Input:** Initial pools $\text{Pool}_1, \text{Pool}_2$ with temperatures $\beta_1, \beta_2$ and pool sizes $N_1, N_2$; LLM evolutionary operator $\mathcal{G}_\phi$;
objective (to minimize) $h(\cdot)$; offspring sizes $o_1, o_2$; parent sizes $\nu_1, \nu_2$; swap size $k$; swap period $t_{\text{swap}}$; total
iterations $T$; powering factor $\xi$; window size $L$; tolerant and threshold for swap rate $\epsilon, \tau$ (e.g., $\epsilon = 0.2, \tau = 0.4$);

**Output:** Final pools $\text{Pool}_1, \text{Pool}_2$;

---

1   $\Gamma \leftarrow []$;
2   **for** $t \leftarrow 1$ **to** $T$ **do**
3     **for** $i \leftarrow 1$ **to** $2$ **do**
      // 1) create mating pool
4       $\mathcal{P} \leftarrow$ `SampleWithoutReplacement`(set=$\text{Pool}_i$, size=$\min(\nu_i, N_i)$, score=$h(\cdot)$, power=$\beta_i$)
      // 2) mate
5       $\mathcal{O} \leftarrow \text{Pool}_i$;
6       **for** $m \leftarrow 1$ **to** $o_i$ **do**
7         Draw parents $x_a, x_b$ uniformly from $\mathcal{P}$;
8         $x_{\text{child}}^{(m)} \leftarrow \mathcal{G}_\phi\big(\text{Prompt}_i, x_a, x_b, h(x_a), h(x_b)\big)$;
9         Append $x_{\text{child}}^{(m)}$ to $\mathcal{O}$;
      // 3) select
10      $\text{Pool}_i \leftarrow$ `SampleWithoutReplacement`(set=$\mathcal{O}$, size=$N_i$, score=$h(\cdot)$, power=$\beta_i$)
    // 4) Swap
11     **if** $t \bmod t_{swap} = 0$ **then**
12       $\gamma \leftarrow$ `PTSwap`($\text{Pool}_1, \text{Pool}_2, \beta_1, \beta_2, N_1, N_2, k, \xi$) ;                // use Algorithm 3
13       Append($\gamma$) to $\Gamma$.
      // 6) adapt Powering factor
14       **if** $length(\Gamma)$ == $L$ **then**
15         **if** $mean(\Gamma) \geq \tau + \epsilon/2$ **then**
16           $\xi \leftarrow \xi * 1.1$;
17         **if** $mean(\Gamma) \leq \tau - \epsilon/2$ **then**
18           $\xi \leftarrow \xi * 0.9$;
19         $\Gamma \leftarrow []$;

20   **return** $\{\text{Pool}_i\}$

---

# B. Additional Experiments and Details

## B.1. Domain-Specific Implementation Conventions

The parallel-tempered evolutionary template described above is shared across all domains, but the concrete implementation differs slightly because the task scores have different numerical meanings and scales, and different tasks inherit different baseline pipelines. The EA loop in all three tasks contains two selection stages—constructing a *mating pool* from the current population, and producing the *next-generation population* by selecting survivors from the union of the current pool and offspring. Following Section 3.2.1, we replace exactly *one* of these two stages with the temperature-weighted stochastic Boltzmann rule (Equation (5)); the other stage remains deterministic. Which stage carries the tempering is chosen on a per-task basis so as to fit naturally with each task's baseline pipeline. We summarize the resulting domain-specific sampling probabilities below; in each formula, $\mathcal{P}_i = \{x_j\}_{j=1}^{N_i}$ denotes the pool at inverse temperature $\beta_i$ (equivalently, temperature $T_i = 1/\beta_i$).

**Molecular discovery (tempering on survivor selection).** For molecular discovery, the mating pool is constructed by sampling candidates with probability proportional to the raw oracle score (no temperature scaling), shared by both pools:

$$p_i^{\text{mating}}(x_j) = \frac{s(x_j) + 0.02/N_i}{\sum_{k=1}^{N_i} \left(s(x_k) + 0.02/N_i\right)}, \tag{11}$$

---

**Algorithm 3:** Swap operator

---

**Input:** Pools $\{\text{Pool}_i\}_{i=1}^2$ with temperatures $\{\beta_i\}_{i=1}^2$ and pool sizes $\{N_i\}_{i=1}^2$; swap size $k$; powering factor $\xi$;
**Output:** Pools $\{\text{Pool}_i\}_{i=1}^2$ after swap
```
// Uniform sample swapping candidates
```
1   $\mathcal{I}_1 \leftarrow$ SampleWithoutReplacement(set=$\text{Pool}_1$, size=$k$, score=$0$, power=$1$);
2   $\mathcal{I}_2 \leftarrow$ SampleWithoutReplacement(set=$\text{Pool}_2$, size=$k$, score=$0$, power=$1$);
3   $\mathcal{R}_1 \leftarrow \text{Pool}_1 \setminus \mathcal{I}_1$;
4   $\mathcal{R}_2 \leftarrow \text{Pool}_2 \setminus \mathcal{I}_2$;
5   **for** $j \leftarrow 1$ **to** $k$ **do**
6      $x_1 \leftarrow \mathcal{I}_1[j], \quad x_2 \leftarrow \mathcal{I}_2[j]$;
7      $\log a \leftarrow -\xi(\beta_1 - \beta_2)(h(x_2) - h(x_1))$;
8      Draw $u \sim \text{Uniform}(0,1)$;
9      **if** $\log u \leq \min(0, \log a)$ **then**
10         $\mathcal{I}_1[j] \leftarrow x_2$;
11         $\mathcal{I}_2[j] \leftarrow x_1$;

12   $\text{Pool}_1 \leftarrow \mathcal{R}_1 \cup \mathcal{I}_1$;
13   $\text{Pool}_2 \leftarrow \mathcal{R}_2 \cup \mathcal{I}_2$;
14   **return** $\{\text{Pool}_i\}_{i=1}^2$

---

where the small additive constant improves numerical stability and prevents candidates with very small scores from receiving exactly zero sampling probability. The temperature is instead applied at the next-generation update step, with sampling probability

$$p_i^{\text{survivor}}(x_j) \; \propto \; s(x_j)^{\beta_i}, \tag{12}$$

which yields sharper selection in the cold pool (large $\beta_i$) and flatter, more exploratory selection in the hot pool (small $\beta_i$).

**Equation discovery and algorithm discovery (tempering on mating-pool sampling).** For these two program-search tasks, the score used for sampling is represented as a log-probability (unnormalized) which we denote by $\ell(x)$, with larger values corresponding to better candidates; the concrete instantiation of $\ell$ is given in the respective task sections below. In both tasks, the temperature is applied at the mating-pool stage, while the next-generation update is deterministic.

For *equation discovery*, we use an effective sampling temperature $0.8\,T_i$ and add a small stabilizing constant:

$$p_i^{\text{mating}}(x_j) = \frac{\exp\left(\ell(x_j)/(0.8\,T_i)\right) + 0.05/N_i}{\sum_{k=1}^{N_i} \left[\exp\left(\ell(x_k)/(0.8\,T_i)\right) + 0.05/N_i\right]}. \tag{13}$$

Due to the wide dynamic range of MSE values, we adopt a different effective energy in the PT swap acceptance criterion; see Section B.3.6 for the ablation.

For *algorithm discovery*, we use the pool temperature directly:

$$p_i^{\text{mating}}(x_j) = \frac{\exp\left(\ell(x_j)/T_i\right)}{\sum_{k=1}^{N_i} \exp\left(\ell(x_k)/T_i\right)}. \tag{14}$$

The next-generation update is identical for the two tasks: each iteration generates a single child candidate that is appended to the current population, and the worst-scoring candidate is removed once the population reaches the capacity of 1000 (otherwise all candidates are retained). This deterministic survivor rule reflects the fact that program evaluation in these domains is expensive, and the population functions as a growing memory buffer of previously discovered hypotheses.

**Parent-pair construction.** After the mating pool is constructed, molecular and algorithm discovery select parent pairs uniformly at random from the mating pool before applying the LLM-based evolutionary operator. In equation discovery, after constructing the mating pool we sort the candidates by score and use the top two candidates as parents.

**Molecular-discovery survivor details.** For molecular discovery, each iteration produces 70 offspring. We concatenate the offspring with the previous population, keep the top 3 candidates deterministically as elites, and fill the remaining population by sampling from the remaining candidates using the tempered survivor probability $p_i^{\text{survivor}}$ defined above. This combines a small amount of elitism with the temperature-weighted survivor selection, preserving high-scoring molecules while still maintaining distinct exploration–exploitation profiles across the cold and hot pools.

### B.1.1. PER-GENERATION BUDGET DETAILS

For completeness and to ensure that the comparisons across methods are made under matched compute budgets, we summarize the per-generation parent and offspring counts for all methods in each of the three domains. For multi-pool methods, the oracle evaluation budget is split across pools so that the total number of oracle calls equals that of the single-pool baselines. The PT swap step itself consumes no additional oracle evaluations. The detailed numbers are given in Appendix Table 4.

*Table 4.* Per-generation parent and offspring counts across all methods and domains. Oracle budgets are shared across pools for multi-pool methods.

| Domain | Method | # Pools | Pop. Size (per pool) | Parents $\nu$ (per prompt) | Offspring $o$ (per pool) | Oracle Budget (total) |
|---|---|---|---|---|---|---|
| Molecular Discovery | MOLLEO | 1 | 120 | 2 | 70 | 10,000 |
| | Ensemble | 2 | 120 | 2 | 70 | 10,000 |
| | Tempering | 1 | 120 | 2 | 70 | 10,000 |
| | EvoDiverse | 2 | 120 | 2 | 70 | 10,000 |
| Equation Discovery | Tempering | 1 | 1000 | 2 | 8 | 1,000 |
| | Ensemble (LLM-SR) | 2 | 1000 | 2 | 8 | 1,000 |
| | EvoDiverse | 2 | 1000 | 2 | 8 | 1,000 |
| Algorithm Discovery | EA | 1 | 1000 | 3 | 1 | 1,000 |
| | Island | 2 | 500 | 3 | 1 | 1,000 |
| | Ensemble | 2 | 500 | 3 | 1 | 1,000 |
| | EvoDiverse | 2 | 500 | 3 | 1 | 1,000 |

## B.2. Molecular Discovery

### B.2.1. TASK SPECIFICATION

**Task description.** In MOLLEO, the JNK3 and GSK3$\beta$ benchmarks treat hit identification as a single-objective black-box search in chemical space. With a fixed oracle evaluation budget, we search molecules with strong predicted inhibitory activity for these targets. Because candidate molecules are modified through discrete operations (e.g., SMILES/graph edits via mutation and crossover) and assessed by a non-differentiable oracle, the objective surface is rugged and prone to local optima, putting pressure on the exploration–exploitation trade-off in evolutionary search.

We study black-box molecular optimization over a discrete search space. Let $\mathcal{M}$ denote the set of candidate molecules represented as SMILES strings, and let

$$\mathcal{M}_{\text{valid}} = \{m \in \mathcal{M} \mid \text{SMILES}(m) \text{ is chemically valid (parsable into a molecular graph)}\}. \tag{15}$$

All optimization is restricted to $\mathcal{M}_{\text{valid}}$; any invalid proposal is rejected and is not counted as an oracle evaluation.

**Oracle and objective.** For each target $t \in \{\text{JNK3}, \text{GSK3}\beta\}$, we query an oracle

$$O_t : \mathcal{M}_{\text{valid}} \to [0, 1], \tag{16}$$

which returns a scalar score. The task is a constrained black-box maximization:

$$m_t^\star \in \arg \max_{m \in \mathcal{M}_{\text{valid}}} O_t(m). \tag{17}$$

To align with the minimization form in Section 2.1, we define

$$h_t(m) := \log(1 - O_t(m)), \tag{18}$$

and equivalently solve $\min_{m \in \mathcal{M}_{\text{valid}}} h_t(m)$.

**Parallel tempered evolutionary search.** Our method maintains two candidate pools $\text{Pool}_1$ and $\text{Pool}_2$ with temperatures (inverse temperatures) $\beta_1 > \beta_2$ and sizes $N_1, N_2$. Within each iteration, each pool performs an EA-style update: (i) *create a mating pool* by sampling $\nu_i$ parents from $\text{Pool}_i$ using a temperature-weighted scheme (`SampleWithoutReplacement`); (ii) *mate* by applying an LLM-based evolutionary operator $\mathcal{G}_\phi$ to pairs of parents, producing $o_i$ offspring proposals; and (iii) *select* the next pool by resampling $N_i$ molecules from the union of current candidates and offspring, again using temperature-weighted sampling. Intuitively, the colder pool (larger $\beta$) concentrates on exploitation, while the hotter pool encourages broader exploration.

**Communication via Metropolis-Hasting swap.** Every $t_{\text{swap}}$ iterations, we perform a PT communication step between the two pools. We uniformly choose $k$ candidates from each pool as swap proposals, and accept pairwise swaps using a Metropolis rule based on the energy difference under $h(\cdot)$:

$$\log a = -\xi(\beta_1 - \beta_2)\left(h(x_2) - h(x_1)\right), \tag{19}$$

where $\xi$ is a powering factor that modulates the swap acceptance rate. This swap exchanges candidates instead of discarding them, allowing promising molecules discovered at the hot temperature to migrate to the cold pool for refinement, while sending some cold-pool candidates to the hot pool to maintain diversity.

**Budget and stopping.** Each run starts from an initial population of 120 molecules sampled from ZINC-250K, and is executed under a fixed oracle-call budget of $B = 10{,}000$. Following MOLLEO, we also use early stopping: the run terminates if the mean score of the current top-100 molecules improves by less than $10^{-3}$ for 5 consecutive epochs. Finally, we adapt $\xi$ based on the empirical swap acceptance rate over a sliding window of length $L$, targeting a desired acceptance level $\tau$ (with tolerance $\epsilon$) as specified in Algorithm 1.

**Diversity-aware selection.** Candidates are first ranked by oracle score in descending order. We then greedily construct the selected set: a molecule is added only if the *average* Tanimoto similarity of its Morgan Fingerprints to those of the already selected molecules is below $0.4$. This process continues no remaining candidates satisfy the threshold. When reporting results on diversity-aware Top-10, we use this method to select the best 10 molecules.

**Diversity-aware Top-10 AUC.** This score measures sample efficiency by computing the area under the curve of the diversity-aware Top-10 average score as a function of the iteration (AUC of "diversity-aware Top-10 average vs. iterations"). This metric therefore captures not only the final solution quality, but also how quickly high-quality *and diverse* candidates are discovered under a fixed evaluation budget.

**Diversity-aware Top-10 Avg.** This score represents the mean oracle score of the diversity-aware Top-10 molecules across the final five iterations prior to convergence. This value reflects the steady state quality of the candidates while maintaining the diversity constraint.

### B.2.2. IMPLEMENTATION DETAILS

**EvoDiverse configuration.** EvoDiverse maintains two concurrently evolving pools, $\text{Pool}_1$ and $\text{Pool}_2$, with inverse temperatures $\beta_1 > \beta_2$. The larger-$\beta$ pool (colder) imposes stronger selection pressure and thus emphasizes exploitation, while the smaller-$\beta$ pool (hotter) yields flatter selection and encourages exploration. We use the following configuration:

- **Pool temperatures.** $\beta_1 = 0.8$ (cold/exploitative) and $\beta_2 = 0.2$ (hot/exploratory).
- **Score used by selection.** We optimize by minimizing the energy $h(m) := \log(1 - O_t(m))$ (equivalently maximizing $O_t$), and all temperature-weighted sampling is defined in terms of $h(\cdot)$.
- **Mating pool and survivor selection.** We follow the two-stage molecular sampling procedure defined in Section B.1: the mating pool is constructed by sampling candidates with probability proportional to the raw oracle score (no temperature, $p_i^{\text{mating}}(m) \propto s(m) + 0.02/N_i$, identical for both pools), while the next-generation survivor selection uses the pool-specific tempered Boltzmann weight $p_i^{\text{survivor}}(m) \propto s(m)^{\beta_i}$. The survivor step retains the top-3 candidates deterministically as elites before tempered `SampleWithoutReplacement` from the union of current candidates and offspring.

- **LLM generation.** We use the same LLM sampling temperature 1.3 in $\mathcal{G}_\phi$ for both pools.
- **Population and budget.** Each run uses an oracle-call budget of $B = 10{,}000$. Each pool generates $o_i = 70$ offspring per iteration (with pool sizes $N_1, N_2$ as specified in experiments).
- **Swap schedule.** We perform PT swap every $t_{\text{swap}} = 5$ iterations. At each swap step, we propose swapping $k$ candidates between the two pools as in Algorithm 3.
- **Adaptive powering factor.** We use a powering factor $\xi$ in the swap acceptance (initialized to $\xi = 2.5$) and adapt it online to maintain a target swap acceptance rate (window size $L$, target $\tau$ with tolerance $\epsilon$), following Algorithm 1.

**PT swap acceptance.** At each swap step, we sample $k$ candidates uniformly from each pool as swap proposals and accept pairwise swaps using a Metropolis rule (cf. Algorithm 3):

$$\log a \;=\; -\xi(\beta_1 - \beta_2)\left(h(x_2) - h(x_1)\right), \tag{20}$$

where $x_1 \in \text{Pool}_1$ and $x_2 \in \text{Pool}_2$ are the proposed swap pair. We accept the swap if

$$\log u \le \min(0, \log a), \quad u \sim \text{Uniform}(0, 1). \tag{21}$$

**MOLLEO configuration (single-pool baseline).** The MOLLEO baseline corresponds to a standard single-population LLM-based evolutionary search without PT swap:

- **Single pool.** One pool with inverse temperature $\beta = 0.8$.
- **No swap.** No swapping/migration is performed.
- **Selection.** The same two-stage molecular sampling procedure as for EvoDiverse (Section B.1) is used: raw-score mating-pool sampling and tempered survivor selection $p^{\text{survivor}}(m) \propto s(m)^\beta$ at $\beta = 0.8$.
- **LLM generation.** Sampling temperature 1.3.
- **Budget and offspring.** Oracle-call budget $B = 10{,}000$; offspring size $o = 70$ per iteration.

This baseline isolates the effect of introducing multiple pools and PT-style communication.

**Ensemble baseline configuration (two independent pools).** To separate the benefit of parallelism from that of communication, we consider two pools evolved independently:

- **Two pools.** Two pools, both with $\beta = 0.8$.
- **No swap.** The pools evolve independently with no information exchange.
- **Selection.** Each pool independently follows the same two-stage molecular sampling procedure as for EvoDiverse (Section B.1): raw-score mating-pool sampling and tempered survivor selection $p^{\text{survivor}}(m) \propto s(m)^\beta$ at $\beta = 0.8$ (identical in both pools).
- **LLM generation.** Both pools use 1.3.
- **Budget and offspring.** Total oracle-call budget $B = 10{,}000$ across both pools; each pool generates $o = 70$ offspring per iteration.
- **Evaluation aggregation.** For reporting, we take the best molecules across the union of candidates evaluated by the two pools.

**Tempering baseline configuration (single hot pool).** Finally, we include a pure-exploration baseline using a single hotter pool:

- **Single pool.** One pool with $\beta = 0.2$.
- **No swap.** No swapping is performed.
- **Selection.** The same two-stage molecular sampling procedure as for EvoDiverse (Section B.1) is used: raw-score mating-pool sampling and tempered survivor selection $p^{\text{survivor}}(m) \propto s(m)^\beta$ at $\beta = 0.2$, yielding a flatter survivor distribution than the MOLLEO/Ensemble baselines and thus more exploratory behavior.
- **LLM generation.** Sampling temperature $t = 1.3$.
- **Budget and offspring.** Oracle-call budget $B = 10{,}000$; offspring size $o = 70$ per iteration.

This baseline tests whether increasing exploration via a higher-temperature selection scheme alone (without multi-pool structure or PT swaps) can account for the improvements of EvoDiverse.

### B.2.3. ADDITIONAL RESULTS AND ANALYSIS

In Appendix Figure 5(a), we report the optimization trajectory comparison between MOLLEO and EvoDiverse on the GSK3$\beta$ task. We observe trends similar to the JNK3 tasks, where EvoDiverse converges to a better score faster than MOLLEO at the cost of a small loss in diversity. In Appendix Figure 5(b), we show the curves used to calculate the results for Table 1. The plots reveal that EvoDiverse converges faster and generally achieves superior scores compared to the baselines. In addition, Appendix Figure 6 shows the optimization trajectories of EvoDiverse molecules relative to the ChEMBL database, where the UMAP projection reveals that optimized molecules remain broadly distributed across chemical space rather than collapsing in a single region. For both tasks, EvoDiverse effectively explores novel regions outside of the existing chemical space.

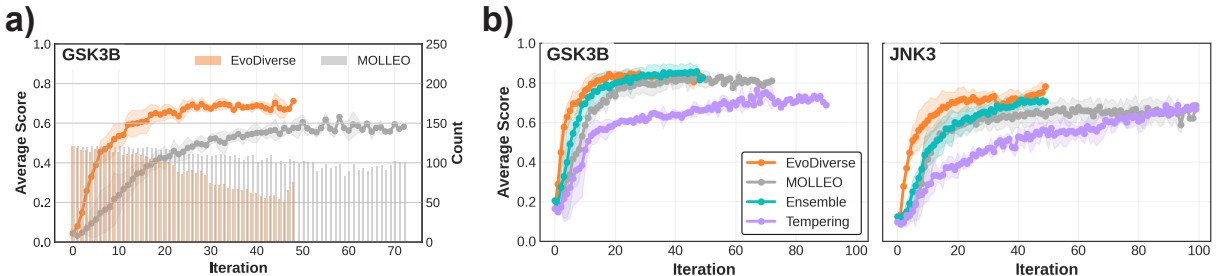

*Figure 5.* Additional results for molecular discovery. (a) Evolution of average GSK3$\beta$ scores (lines) and the corresponding number of samples (bars) passing diversity-aware selection for MOLLEO and EvoDiverse. (b) Optimization trajectories of average GSK3$\beta$ and JNK3 scores compared against baselines. Average scores are calculated based on diversity-aware Top-10 molecules from each iteration.

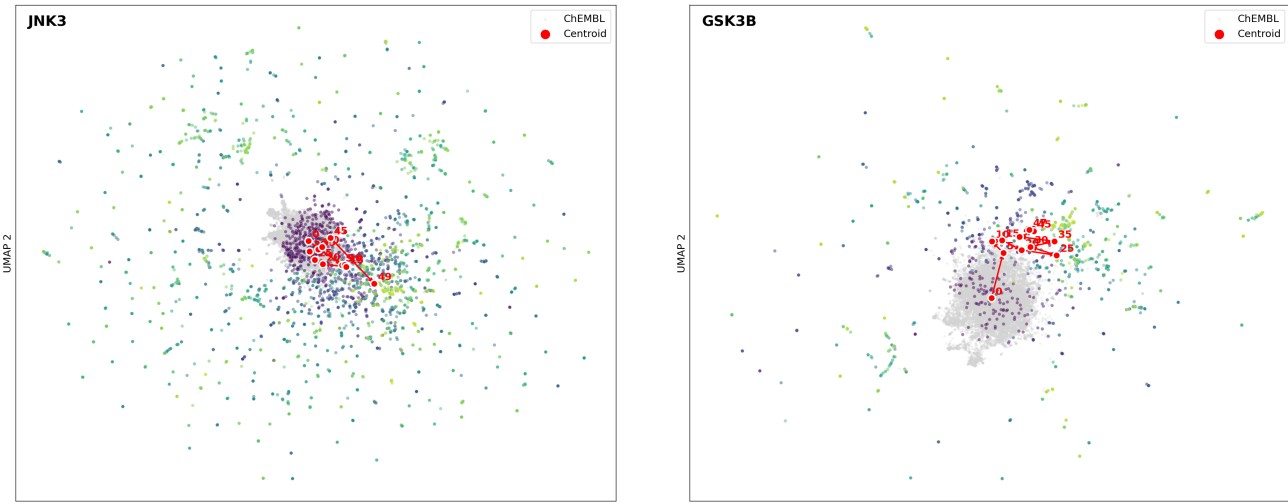

*Figure 6.* Trajectory of explored chemical space for EvoDiverse during optimization for JNK3 and GSK3$\beta$ scores. Gray dots represent a UMAP projection of Morgan fingerprints for all ChEMBL 36 data. Red dots indicate the centroids of molecules optimized by EvoDiverse from each iteration, illustrating an evolving trajectory through chemical space. For all colored points, increasing brightness denotes later iterations. These results demonstrate that EvoDiverse starts from known chemistry and gradually explore novel regions outside existing chemical space through iterations.

Besides evaluating LLM-based optimization alone, we also test whether EvoDiverse can enhance classical algorithms such as GraphGA (Jensen, 2019). In Appendix Table 5 and Appendix Figure 7(b), we demonstrate the superior performance of EvoDiverse in converging more rapidly to high-scoring samples compared to baseline methods. Furthermore, we provide the same analysis regarding QED and SA score filtering and observe a consistent trend as the LLM-based results (Appendix Figure 7(a)). For molecules that pass both filters, the average score of candidates proposed by EvoDiverse remains superior to the vanilla GraphGA baseline.

*Table 5.* Additional results for molecular discovery comparing EvoDiverse against GraphGA. Here, EvoDiverse is an adapted version of the GraphGA method, distinct from EvoDiverse in Main which uses LLM for optimization. The reported metrics are the same as Table 1.

| Target | Method | Metrics | |
| --- | --- | --- | --- |
| | | Top-10 AUC ↑ | Top-10 Avg ↑ |
| JNK3 | GraphGA | $0.35 \pm 0.04$ | $\mathbf{0.59 \pm 0.10}$ |
| | EvoDiverse | $\mathbf{0.43 \pm 0.07}$ | $0.52 \pm 0.06$ |
| GSK3$\beta$ | GraphGA | $0.59 \pm 0.04$ | $0.72 \pm 0.06$ |
| | EvoDiverse | $\mathbf{0.67 \pm 0.02}$ | $\mathbf{0.80 \pm 0.06}$ |

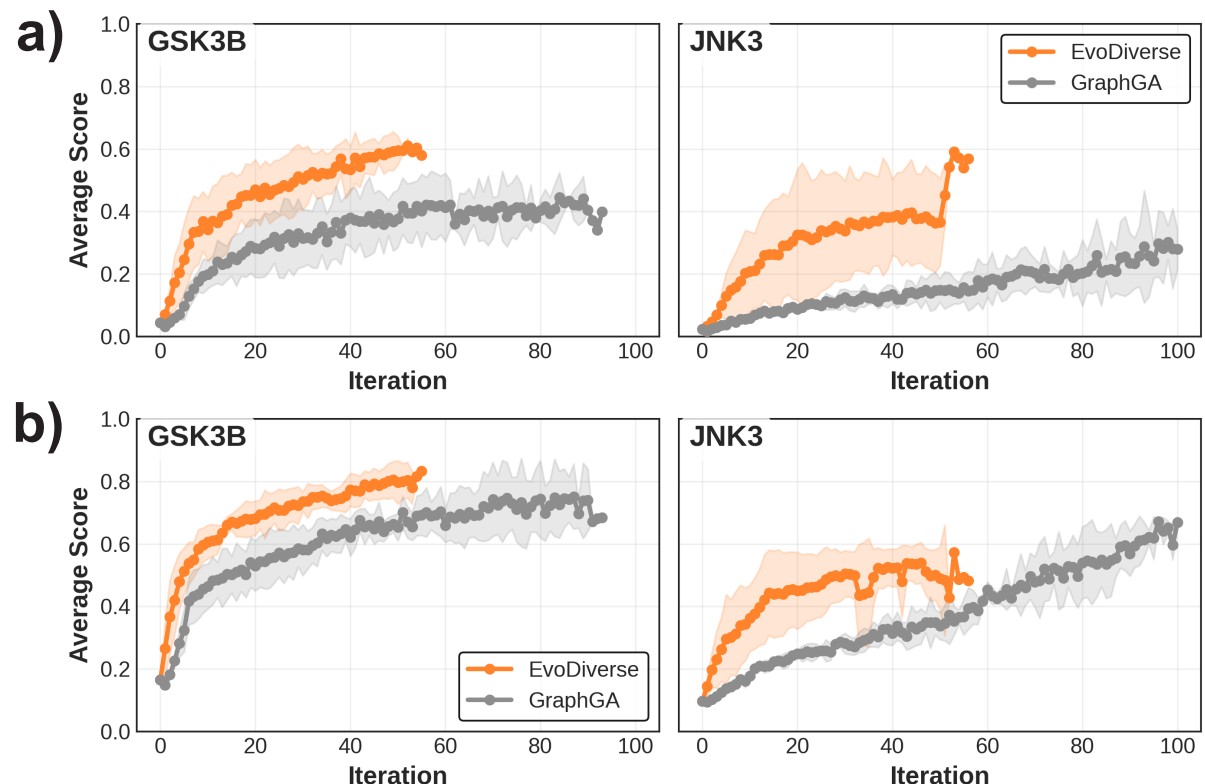

*Figure 7.* Optimization trajectories of average GSK3$\beta$ and JNK3 scores of EvoDiverse compared against GraphGA. Here, EvoDiverse is an adapted version of the GraphGA method, distinct from EvoDiverse in Main which uses LLM for optimization. (a) Average scores are calculated based on all molecules from each iteration satisfying QED > 0.5 and SA < 5.5. (b) Average scores are calculated based on diversity-aware Top-10 molecules from each iteration. EvoDiverse (green) shows accelerated convergence and higher average scores across both selection criteria in most cases.

### B.2.4. ABLATION: METROPOLIS-HASTINGS SWAP

To isolate the role of the MH acceptance gate in the PT swap, we compare EvoDiverse against **Ensemble** (no swap) and **Random Swap** (the identical two-pool architecture but with the MH gate disabled. All proposed swaps are accepted unconditionally, equivalent to $\xi=0$). Results on the JNK3 and GSK3$\beta$ benchmarks are reported in Appendix Table 6. Both Random Swap and EvoDiverse outperform the swap-free Ensemble baseline, confirming that inter-pool communication is beneficial. However, the MH gate further reduces variance and improves quality. For example, on JNK3 the Top-10 AUC standard deviation drops from 0.08 (Random Swap) to 0.05 (EvoDiverse); the MH criterion selectively transfers candidates consistent with both temperatures, preventing the disruptive injection of incompatible programs into the cold pool.

*Table 6.* MH swap ablation on molecular discovery. "Random Swap" uses the identical two-pool architecture but accepts all proposed swaps unconditionally (no MH acceptance gate). Results averaged over 3 seeds.

| Target | Method | Top-10 AUC ↑ | Top-10 Avg ↑ |
|---|---|---|---|
| | Ensemble (no swap) | $0.54 \pm 0.04$ | $0.71 \pm 0.03$ |
| JNK3 | Random Swap (no MH gate) | $0.62 \pm 0.08$ | $0.68 \pm 0.09$ |
| | EvoDiverse (MH swap) | $\mathbf{0.63} \pm 0.05$ | $\mathbf{0.74} \pm 0.04$ |
| | Ensemble (no swap) | $0.73 \pm 0.04$ | $\mathbf{0.85} \pm 0.04$ |
| GSK3$\beta$ | Random Swap (no MH gate) | $\mathbf{0.76} \pm 0.02$ | $\mathbf{0.85} \pm 0.03$ |
| | EvoDiverse (MH swap) | $\mathbf{0.76} \pm 0.02$ | $0.82 \pm 0.03$ |

### B.2.5. ABLATION: SAME PROMPT ACROSS POOLS

By default, EvoDiverse uses pool-specific LLM prompts: the cold pool is asked to refine known high-performing motifs, and the hot pool to propose structurally diverse candidates. To verify that the improvements over the MOLLEO baseline are driven by the PT mechanism itself rather than by prompt engineering, we run EvoDiverse with the cold-pool prompt used in *both* pools, so that the only inter-pool difference is the selection inverse temperature $\beta$. Appendix Table 7 shows that, even under identical prompts, EvoDiverse retains substantial gains over MOLLEO: $+12\%$ Top-10 AUC on JNK3 and $+19\%$ on GSK3$\beta$. This confirms that pool-specific prompting is a complementary enhancement and not the source of the framework-level gains.

*Table 7.* Same-prompt ablation on molecular discovery. "EvoDiverse (same prompt)" uses the cold-pool prompt for *both* pools; the only inter-pool difference is the selection inverse temperature $\beta$. Results averaged over 3 seeds.

| Target | Method | Top-10 AUC ↑ | Top-10 Avg ↑ |
|---|---|---|---|
| | MOLLEO (single pool) | $0.58 \pm 0.05$ | $0.66 \pm 0.06$ |
| JNK3 | EvoDiverse (same prompt) | $\mathbf{0.65} \pm 0.05$ | $0.73 \pm 0.04$ |
| | EvoDiverse (different prompts) | $0.63 \pm 0.05$ | $\mathbf{0.74} \pm 0.04$ |
| | MOLLEO (single pool) | $0.70 \pm 0.03$ | $0.82 \pm 0.03$ |
| GSK3$\beta$ | EvoDiverse (same prompt) | $\mathbf{0.83} \pm 0.08$ | $\mathbf{0.89} \pm 0.07$ |
| | EvoDiverse (different prompts) | $0.76 \pm 0.02$ | $0.82 \pm 0.03$ |

### B.2.6. COMPUTATIONAL COST

All methods on molecular discovery share the same oracle budget of 10,000 evaluations. Appendix Table 8 and 9 report the LLM call counts, token consumption, wall-clock time, and invalid-proposal rate. EvoDiverse uses LLM-call and token resources comparable to the baselines, and its wall-clock time is similar to or shorter than MOLLEO's because the two pools execute in parallel. The MOLLEO baseline uses fewer calls on GSK3$\beta$ only because it triggers the MOLLEO early-stopping criterion at a sub-optimal solution. Invalid-proposal rates are uniformly low across all methods ($< 1.2\%$), and EvoDiverse exhibits the lowest rate ($0.4$–$0.6\%$). Overall, *the diversity and quality gains of EvoDiverse come from better allocation of a fixed evaluation budget, not from additional computation.*

*Table 8.* Computational cost comparison on JNK3 molecular discovery. All methods share the same oracle budget of 10,000. Results averaged over 3 seeds. For EvoDiverse, "(same)" means we use the same prompt for both pools and "(diff)" means we use different prompts for different pools.

| Method | LLM Calls | Total Tokens (M) | Wall-Clock (h) | Invalid Rate (%) | Top-10 AUC ↑ |
|---|---|---|---|---|---|
| MOLLEO | 9300 | 8.02 | 4.06 | 0.7 | 0.58 |
| Ensemble | 10068 | 8.56 | 3.87 | 1.0 | 0.54 |
| Tempering | 9842 | 9.10 | 3.96 | 1.2 | 0.46 |
| EvoDiverse (diff) | 9813 | 9.39 | 3.82 | 0.6 | 0.63 |
| Random Swap | 10280 | 10.05 | 4.40 | 0.6 | 0.62 |
| EvoDiverse (same) | 10187 | 8.15 | 3.84 | 0.4 | **0.65** |

*Table 9.* Computational cost comparison on GSK3$\beta$ molecular discovery. Format identical to Appendix Table 8.

| Method | LLM Calls | Total Tokens (M) | Wall-Clock (h) | Invalid Rate (%) | Top-10 AUC ↑ |
|---|---|---|---|---|---|
| MOLLEO | 3220 | 2.76 | 2.24 | 0.6 | 0.70 |
| Ensemble | 6180 | 5.69 | 3.68 | 0.6 | 0.73 |
| Tempering | 9834 | 9.56 | 4.26 | 0.9 | 0.58 |
| EvoDiverse (diff) | 8673 | 8.43 | 3.44 | 0.6 | 0.76 |
| Random Swap | 10187 | 10.27 | 4.15 | 0.6 | 0.76 |
| EvoDiverse (same) | 5507 | 4.72 | 3.12 | 0.5 | **0.83** |

### B.2.7. NON–DIVERSITY-AWARE TOP-$k$ SCORES

The main paper reports diversity-aware Top-$k$ scores that filter candidates by structural dissimilarity. For completeness, Appendix Table 10 reports the analogous *standard* Top-$k$ scores ranked purely by oracle score with no diversity filtering. EvoDiverse achieves the best Top-10 and Top-100 average scores on both targets, confirming that the gains in diversity-aware metrics do not come at the cost of raw score quality.

*Table 10.* Standard (non-diversity-aware) evaluation for molecular discovery. Scores are the mean oracle scores of the top-$k$ molecules from the last iteration of each method ranked purely by score, without any diversity filtering. Results averaged over 3 seeds.

| Target | Method | Std Top-10 Avg ↑ | Std Top-100 Avg ↑ |
|---|---|---|---|
| JNK3 | MOLLEO | $0.79 \pm 0.02$ | $0.62 \pm 0.02$ |
| | Ensemble | $0.75 \pm 0.02$ | $0.66 \pm 0.03$ |
| | Tempering | $0.63 \pm 0.05$ | $0.44 \pm 0.06$ |
| | EvoDiverse | $\mathbf{0.82} \pm 0.03$ | $\mathbf{0.69} \pm 0.05$ |
| GSK3$\beta$ | MOLLEO | $0.87 \pm 0.02$ | $0.71 \pm 0.02$ |
| | Ensemble | $0.88 \pm 0.03$ | $0.79 \pm 0.06$ |
| | Tempering | $0.75 \pm 0.11$ | $0.54 \pm 0.06$ |
| | EvoDiverse | $\mathbf{0.92} \pm 0.02$ | $\mathbf{0.81} \pm 0.03$ |

In addition, Appendix Table 11 reports the last-iteration diversity-aware Top-10 set together with secondary chemical-quality metrics: molecular and scaffold diversity, the number of unique Murcko scaffolds, drug-likeness (QED), and synthetic accessibility (SA). EvoDiverse achieves the strongest balance of high oracle score, high QED, and high SA, with no sign of reward hacking.

### B.2.8. SCAFFOLD-LEVEL DIVERSITY AND UNIDOCK DOCKING

To probe whether EvoDiverse maintains diversity at the scaffold level (rather than only at the full-molecule fingerprint level), we evaluate three diversity-aware selection variants on the diversity-aware Top-10 set from the last iteration, summarized

*Table 11.* Last-iteration top-10 molecules from diversity-aware selection (average Tanimoto similarity to previously selected molecules on full-molecule Morgan fingerprints required to be below 0.4). We report mean oracle score, molecular diversity ($\text{mean}_{i<j}(1-T_{ij})$ on full-molecule fingerprints), the number of unique Murcko scaffolds, scaffold diversity (the same on Murcko-scaffold fingerprints), and QED and SA computed via RDKit. Results averaged over 3 seeds.

| Target | Method | Top-10 Avg Score ↑ | Molecular Diversity ↑ | Unique Scaffolds ↑ | Scaffold Diversity ↑ | QED ↑ | SA ↑ |
|---|---|---|---|---|---|---|---|
| JNK3 | MOLLEO | $0.68 \pm 0.04$ | $0.63 \pm 0.00$ | $10.00 \pm 0.00$ | $0.53 \pm 0.03$ | $0.17 \pm 0.09$ | $0.70 \pm 0.08$ |
| | Ensemble | $0.72 \pm 0.02$ | $0.64 \pm 0.00$ | $9.67 \pm 0.58$ | $0.59 \pm 0.02$ | $0.12 \pm 0.05$ | $0.71 \pm 0.04$ |
| | Tempering | $0.59 \pm 0.08$ | $0.65 \pm 0.00$ | $10.00 \pm 0.00$ | $0.59 \pm 0.03$ | $0.12 \pm 0.00$ | $0.69 \pm 0.02$ |
| | EvoDiverse | $0.74 \pm 0.04$ | $0.64 \pm 0.01$ | $9.67 \pm 0.58$ | $0.54 \pm 0.06$ | $0.21 \pm 0.10$ | $0.75 \pm 0.05$ |
| GSK3$\beta$ | MOLLEO | $0.83 \pm 0.03$ | $0.64 \pm 0.01$ | $7.67 \pm 2.52$ | $0.42 \pm 0.17$ | $0.40 \pm 0.08$ | $0.80 \pm 0.03$ |
| | Ensemble | $0.85 \pm 0.04$ | $0.65 \pm 0.03$ | $7.33 \pm 1.15$ | $0.47 \pm 0.01$ | $0.39 \pm 0.15$ | $0.77 \pm 0.03$ |
| | Tempering | $0.70 \pm 0.06$ | $0.66 \pm 0.01$ | $10.00 \pm 0.00$ | $0.62 \pm 0.03$ | $0.21 \pm 0.09$ | $0.73 \pm 0.03$ |
| | EvoDiverse | $0.82 \pm 0.05$ | $0.63 \pm 0.02$ | $9.67 \pm 0.58$ | $0.50 \pm 0.01$ | $0.45 \pm 0.19$ | $0.78 \pm 0.04$ |

in Appendix Table 12. At convergence, all methods produce at least 10 diverse molecules with distinct scaffolds, and EvoDiverse maintains both a high average oracle score and high scaffold diversity simultaneously on both targets.

*Table 12.* Diversity-aware Top-10 candidate sets at the last iteration. MOL DIV TOP-10 AVERAGE SCORE is the mean score of up to ten last-iteration molecules chosen by diversity-aware selection on full-molecule Morgan fingerprints. UNIQUE SCAF. TOP-10 AVG. SCORE is the mean score of up to ten molecules taken in score order with distinct Murcko scaffolds. SCAF. DIV. TOP-10 AVG. SCORE is the mean score of up to ten molecules chosen by the same diversity rule using Murcko-scaffold Morgan fingerprints. Results averaged over 3 seeds.

| Target | Method | Mol Div Top-10 Average Score ↑ | Unique Scaf. Top-10 Avg. Score ↑ | Scaf. Div. Top-10 Avg. Score ↑ |
|---|---|---|---|---|
| JNK3 | MOLLEO | $0.68 \pm 0.04$ | $0.78 \pm 0.02$ | $0.59 \pm 0.07$ |
| | Ensemble | $0.72 \pm 0.02$ | $0.74 \pm 0.02$ | $\mathbf{0.70 \pm 0.02}$ |
| | Tempering | $0.59 \pm 0.08$ | $0.63 \pm 0.05$ | $0.58 \pm 0.08$ |
| | EvoDiverse | $\mathbf{0.74 \pm 0.04}$ | $\mathbf{0.79 \pm 0.03}$ | $0.60 \pm 0.21$ |
| GSK3$\beta$ | MOLLEO | $0.83 \pm 0.03$ | $0.85 \pm 0.04$ | $0.75 \pm 0.06$ |
| | Ensemble | $\mathbf{0.85 \pm 0.04}$ | $0.86 \pm 0.04$ | $\mathbf{0.81 \pm 0.03}$ |
| | Tempering | $0.70 \pm 0.06$ | $0.75 \pm 0.11$ | $0.67 \pm 0.04$ |
| | EvoDiverse | $0.82 \pm 0.05$ | $\mathbf{0.90 \pm 0.03}$ | $0.67 \pm 0.19$ |

To further validate the candidates beyond the surrogate oracle, we perform UniDock docking on the diversity-selected Top-10 molecules. Appendix Table 13 reports docking scores (kcal/mol; lower is better) for both the very last iteration and the last 5 iterations. EvoDiverse achieves the best JNK3 docking score in both cases and the best GSK3$\beta$ score under the last-5-iterations average, while also showing the lowest variance, indicating that the discovered molecules generalize to physics-based binding scoring beyond the surrogate oracle.

### B.2.9. STATISTICAL SIGNIFICANCE

Appendix Table 14 reports paired one-sided $t$-tests of EvoDiverse over each baseline on molecular discovery ($N = 3$ seeds). On JNK3, the Top-10 AUC improvement of EvoDiverse is significant over MOLLEO ($p = 0.011$) and Tempering ($p = 0.006$); the comparison with Ensemble is inconclusive given the small sample size ($p = 0.111$). However, the secondary chemical-quality analysis in Appendix Table 11 suggests the Ensemble baseline achieves a lower QED on JNK3 (0.12 vs. 0.21), indicating possible reward hacking that the AUC alone does not capture. On GSK3$\beta$, the improvement over Tempering is highly significant ($p = 5.4 \times 10^{-6}$).

*Table 13.* UniDock docking on diversity-selected top-10 molecules for the last few iterations. The docking score (kcal/mol) results are averaged over 3 seeds.

| Target | Method | Last iter. docking score ↓ | Last 5 iters. docking score ↓ |
|--------|--------|----------------------------|-------------------------------|
| JNK3 | MOLLEO | $-7.53 \pm 0.80$ | $-7.32 \pm 0.77$ |
| | Ensemble | $-6.77 \pm 1.32$ | $-7.07 \pm 1.07$ |
| | Tempering | $-6.51 \pm 0.67$ | $-6.37 \pm 0.79$ |
| | EvoDiverse | $\mathbf{-7.83} \pm 0.08$ | $\mathbf{-7.75} \pm 0.33$ |
| GSK3$\beta$ | MOLLEO | $-8.89 \pm 0.20$ | $-9.08 \pm 0.05$ |
| | Ensemble | $\mathbf{-9.37} \pm 0.52$ | $-9.07 \pm 0.65$ |
| | Tempering | $-8.95 \pm 0.37$ | $-8.93 \pm 0.44$ |
| | EvoDiverse | $-9.10 \pm 0.38$ | $\mathbf{-9.19} \pm 0.25$ |

*Table 14.* Statistical significance of EvoDiverse over baselines on molecular discovery (paired $t$-test, $N = 3$ seeds, one-sided).

| Target | Comparison | Top-10 AUC $p$-value | Top-10 Avg $p$-value |
|--------|-----------|----------------------|----------------------|
| JNK3 | vs. MOLLEO | 0.011 | 0.035 |
| | vs. Ensemble | 0.111 | 0.261 |
| | vs. Tempering | 0.006 | 0.020 |
| GSK3$\beta$ | vs. MOLLEO | 0.086 | 0.635 |
| | vs. Ensemble | 0.132 | 0.815 |
| | vs. Tempering | $5.40e-06$ | 0.012 |

### B.2.10. DIAGNOSTIC CURVES

Appendix Figure 8 plots the adaptive powering factor $\xi$ and the empirical swap acceptance rate over iterations for EvoDiverse on both molecular discovery targets. The acceptance rate quickly stabilises near 30–40%, and $\xi$ increases monotonically as the underlying score distributions sharpen during search. The qualitative behaviour matches that observed for equation and algorithm discovery (Appendix Figure 11 and 14).

### B.3. Equation Discovery

#### B.3.1. TASK SPECIFICATION

Scientific equation discovery (symbolic regression) seeks to infer the underlying physical or biological laws of a system in the form of concise, interpretable mathematical expressions. Given observations $\{(x_i, y_i)\}_{i=1}^{N}$, where $x_i \in \mathbb{R}^d$ represents system variables and $y_i \in \mathbb{R}$ the measured response, the objective is to recover a symbolic function $\hat{f}(x)$ that both fits the data and extrapolates beyond the observed regime. Unlike black-box regression, equation discovery is inherently a structured search problem over a discrete space of mathematical programs. Successful solutions must jointly satisfy numerical fidelity, symbolic simplicity, and semantic validity, closely mirroring the process of scientific hypothesis formation. This makes symbolic regression a natural benchmark for evaluating LLM-driven evolutionary discovery methods that combine program synthesis with iterative refinement. Our framework casts hypothesis generation as symbolic program synthesis. At the start of the search, the LLM is prompted with a brief task description, variable definitions, and a small set of elementary examples (e.g., linear or low-degree polynomial relations), which initialize an experience buffer. At each iteration, the LLM proposes candidate equation templates expressed as executable Python functions, specifying symbolic structure while leaving continuous coefficients as free parameters. Each candidate template is instantiated by fitting its parameters to data using standard numerical optimization (e.g., BFGS). The resulting equation is then evaluated by executing the program and measuring prediction error via negative mean squared error (MSE). Invalid, unstable, or degenerate programs are filtered out, while high-quality hypotheses are retained in a structured, multi-pool buffer that explicitly promotes diversity and prevents early collapse of the search. The search proceeds iteratively by conditioning subsequent LLM proposals on selected hypotheses from this buffer, allowing the system to progressively refine promising symbolic structures while continuing to explore alternative formulations. After a fixed computational budget (up to 1000 iterations in our experiments), the

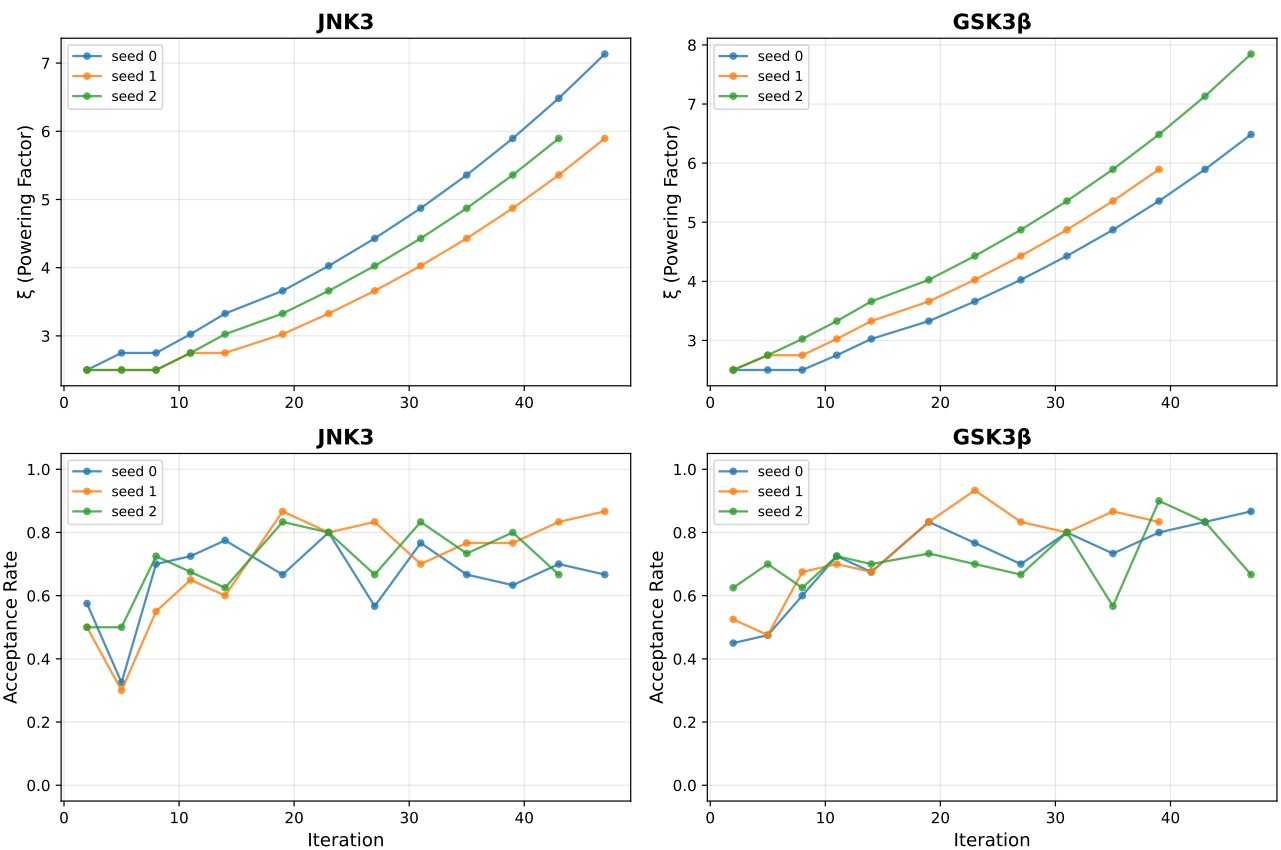

*Figure 8.* Diagnostic curves for EvoDiverse on molecular discovery. The adaptive powering factor $\xi$ (top) and the swap acceptance rate (bottom).

highest-scoring equation is selected as the discovered governing law.

We evaluate this process on LLM-SRBench ([Shojaee et al., 2025b](#)), a diverse benchmark spanning physics, biology, chemistry, and materials science. Each task provides standardized train/test splits. All data are normalized following standard practice, and identical splits are used across methods for fair comparison. We experiment with both open and closed LLM backbones, including `deepseek-v3.2` and `gpt-5`, as hypothesis generators. All LLM-based baselines operate under the same operator grammar, prompt structure, evaluation protocol, and iteration budget; differences arise solely from how candidate populations are organized and reused across iterations. Results are averaged over multiple random seeds and aggregated across datasets.

We evaluate solution quality using both accuracy-based and error-based metrics to capture complementary aspects of symbolic discovery. In particular, we report an accuracy measure defined by a relative error tolerance $\tau = 0.1$, which enforces a stringent notion of fidelity to the data:

$$\text{Acc}_\tau = \mathbb{I}\left(\max_{1 \le i \le N_{\text{test}}} \frac{|\hat{y}_i - y_i|}{|y_i|} \le \tau\right), \tag{22}$$

where $\hat{y}_i$ and $y_i$ denote the predicted and ground-truth outputs, respectively, and $\mathbb{I}(\cdot)$ is the indicator function. This metric requires all test points to satisfy the relative error constraint.

We additionally report the normalized mean squared error (NMSE),

$$\text{NMSE} = \frac{\sum_{i=1}^{N_{\text{test}}} (\hat{y}_i - y_i)^2}{\sum_{i=1}^{N_{\text{test}}} (y_i - \bar{y})^2}, \tag{23}$$

where $\bar{y}$ denotes the mean of the ground-truth outputs, providing a comp

### B.3.2. IMPLEMENTATION DETAILS

**EvoDiverse Configuration.** Our EvoDiverse follows the parallel-tempered evolutionary template in Algorithms 1 and 3 and maintains two concurrently evolving pools, $\text{Pool}_1$ (cold) and $\text{Pool}_2$ (hot), with inverse temperatures $\beta_1 > \beta_2$. For the SR task, each candidate program induces an equation and is scored by its mean squared error (MSE) on the dataset; we therefore define the energy to minimize as

$$h(x) := \text{MSE}(x), \tag{24}$$

and all temperature-weighted sampling and PT swaps are expressed in terms of $h(\cdot)$.

The key configuration choices are:

- **Pool temperatures.** Cold pool $\beta_1 = 0.8$ and hot pool $\beta_2 = 0.01$.
- **Selection (parents and survivors).** Within $\text{Pool}_i$, we sample parents (and resample survivors) using Boltzmann weights following Section B.1.
- **LLM generation.** We use different decoding settings for the two pools to further bias exploration/exploitation: the cold pool uses temperature 0.7 with top-$p = 0.8$, while the hot pool uses temperature 1.0 with top-$p = 0.95$.
- **Pool capacity and evaluation budget.** Each pool maintains up to $N_i = 1000$ program candidates. We impose a total evaluation budget of $B = 1000$ oracle calls across both pools (i.e., at most $B$ distinct programs are evaluated by the MSE oracle in a run).
- **Swap schedule.** Every $t_{\text{swap}} = 5$ iterations, we propose swaps between the two pools using the PT swap operator in Algorithm 3 (swap size $k$ as specified in experiments).
- **Adaptive powering factor.** We initialize $\xi = 2.5$ and adapt $\xi$ online to maintain a target swap acceptance rate of $\tau = 0.3$ (with multiplicative updates $\xi \leftarrow 1.1\xi$ or $\xi \leftarrow 0.9\xi$ following Algorithm 1).

**PT swap acceptance.** At each swap step, for a proposed pair $x_1 \in \text{Pool}_1$ and $x_2 \in \text{Pool}_2$, we compute the Metropolis log acceptance ratio as

$$\log a = -\xi(\beta_1 - \beta_2)\big(\log h(x_2) - \log h(x_1)\big), \tag{25}$$

and accept the swap if $\log u \leq \min(0, \log a)$ with $u \sim \text{Uniform}(0, 1)$. Note this swap rate is slightly different than the one obtained by taking $h$ as the energy. In Section B.3.6, we explain this choice and provide ablation to justify it.

This communication step selectively transfers low-energy (low-MSE) programs discovered under the hot dynamics into the cold pool for refinement, while allowing some cold-pool candidates to move to the hot pool to escape stagnation.

**Ensemble (LLM-SR) Configuration.** To separate the benefit of parallelism from that of communication, we consider an ensemble baseline with two independent pools and no swapping:

- **Two pools.** Two pools, both with $\beta = 1.0$.
- **No swap.** No swapping/migration is performed (equivalently, $t_{\text{swap}}$ is set to a very large value).
- **Selection (parents and survivors).** We sample parents (and resample survivors) using Boltzmann weights following Section B.1.
- **LLM generation.** Both pools use the same decoding configuration: temperature 0.7 with top-$p = 0.8$.
- **Pool capacity and budget.** Each pool maintains up to 1000 candidates; the total evaluation budget is $B = 1000$ across both pools (split evenly unless stated otherwise).

This baseline corresponds to two independent evolutionary searches without information exchange; improvements can only arise from running multiple searches in parallel rather than from PT-style communication.

**Tempering Configuration (single hot pool).** Finally, we include a single-pool baseline that uses a hotter/low-selection-pressure setting:

- **Single pool.** One pool with $\beta = 0.01$.
- **No swap.** No swapping is applicable.
- **Selection (parents and survivors).** We sample parents (and resample survivors) using Boltzmann weights following Section B.1, yielding flatter selection pressure due to the higher temperature adopted.
- **LLM generation.** Temperature 1.0 with top-$p = 0.95$.
- **Pool capacity and budget.** Pool capacity 1000; evaluation budget $B = 1000$.

This baseline tests whether increased exploration alone (without the two-pool structure and PT swaps) can account for the gains of EvoDiverse.

### B.3.3. ADDITIONAL DETAILS ON SWAP DYNAMICS IN EVODIVERSE AND ADAPTIVE SCALING

The parallel tempering swap mechanism in EvoDiverse enables selective information transfer between pools. Over the course of 1000 evaluations, the system attempts swaps approximately every 5 iterations (migration interval), with each swap attempt involving sampling one candidate from each pool using softmax selection probabilities and applying the Metropolis-Hastings acceptance criterion. The adaptive $\xi$ mechanism successfully calibrates the swap barrier to the problem's fitness scale. Starting from $\xi = 2.5$, the system automatically adjusts to maintain a target swap acceptance rate (typically around 30% $\pm$ 10%). This adaptation ensures consistent information flow between pools regardless of the absolute magnitude of fitness scores. The relatively moderate acceptance rate indicates healthy thermodynamic communication: swaps are selective (not all attempts succeed) but frequent enough to enable effective information transfer between pools. The total number of programs exchanged represents a modest but crucial flow of genetic material between pools—far less than would occur with mass and random migration strategies, yet more effective at maintaining diversity while enabling convergence. This selective exchange enables the designed "funnel" behavior: the Hot pool discovers diverse candidates through exploration, while the Cold pool refines promising solutions through exploitation, with the swap mechanism enabling productive candidates to flow from Hot to Cold while allowing stuck solutions to escape from Cold to Hot.

### B.3.4. DIVERSITY EVALUATION

In equation discovery, we quantify diversity of equation program candidates in buffer based on the diversity quantified within each pool as the mean pairwise embedding distance computed using CodeBERT (Feng, 2020) embeddings. For a pool containing $n$ programs with embeddings $\{e_1, e_2, \ldots, e_n\}$, we compute the diversity metric as: Diversity $= \frac{1}{\binom{n}{2}} \sum_{i<j} d_{\text{cosine}}(e_i, e_j)$ where $d_{\text{cosine}}$ is the cosine distance between embeddings. CodeBERT embeddings are obtained by tokenizing each program's code body, passing it through the pre-trained CodeBERT model, and computing a mean-pooled representation over the sequence of hidden states weighted by attention masks. This semantic diversity metric captures structural and functional differences between symbolic programs, enabling us to evaluate how effectively each method maintains exploration of the solution space. In our experiments, we track diversity metrics for:

- **Overall buffer**: All programs across all pools. Diversity reported for programs within each individual pool (Pool 0 / Pool 1) and average among these within pool measures.
- **Top-100**: The 100 highest-scoring programs (per-pool and overall)

Table 15 provides comprehensive diversity analysis across baseline experiments on datasets from different domains in LLM-SRBench, and two LLM backbones (deepseek-v3.2 and GPT-5). The diversity is reported based on per-pool break downs (and the average of these within pool diversity measures among two pools) over the whole buffer as well as the top-100 elite set. The results demonstrate that EvoDiverse achieves productive pool differentiation with asymmetric elite contributions, while Ensemble on average faces with more isolation between pools among the top-100 candidates. We also observe that EvoDiverse helps to significantly boost diversity across all the experiments.

The diversity landscape visualization (shown in Figure 10) reveals distinct population structures across methods. EvoDiverse shows two partially overlapping clusters: the Cold pool forms clusters in high-score regions, while the Hot pool explores a broader area of the solution space. The overlap region represents solutions that successfully transferred between pools via thermodynamic swaps. Ensemble baseline displays two disconnected clusters with no overlap, and Tempering exhibits a single cluster with moderate spread, representing standard evolutionary dynamics but lacking the structured specialization of EvoDiverse's dual-pool architecture.

### B.3.5. ADDITIONAL RESULTS

Appendix Figure 9 provides an extended comparison of EvoDiverse with representative LLM-based baselines (Tempering and Ensemble/LLM-SR (Shojaee et al., 2025a)) and the state-of-the-art non-LLM symbolic regression method PySR (Cranmer, 2023) across the four LLM-SRBench (Shojaee et al., 2025b) domains. Results for LLM-based methods are averaged over DeepSeek-V3.2 and GPT-5 backbones to highlight method-level trends independent of backbone choice. Overall, results show that EvoDiverse consistently achieves lower NMSE and higher $\text{Acc}_{0.1}$ than both LLM-based baselines, indicating improved optimization quality and discovery reliability. Compared to PySR, which struggles in more complex

*Table 15.* Detailed diversity comparison of EvoDiverse and baselines on equation discovery. Results show all-buffer and Top-100 diversity across LLM-SRBench domains for DeepSeek-V3.2 and GPT-5 backbones, with hot/cold pool breakdowns (overall in parentheses)

| Dataset | Method | DeepSeek-V3.2 | | GPT-5 | |
|---|---|---|---|---|---|
| | | All Buffer Diversity ↑ (Hot/Cold) | Top-100 Diversity ↑ (Hot/Cold) | All Buffer Diversity ↑ (Hot/Cold) | Top-100 Diversity ↑ (Hot/Cold) |
| Physics | Tempering | **0.308** | **0.303** | 0.280 | 0.268 |
| | Ensemble (LLM-SR) | 0.293/0.285 (0.289) | 0.296/0.281 (0.288) | 0.293/0.275 (0.284) | 0.290/0.266 (0.278) |
| | EvoDiverse | 0.324/0.292 (**0.308**) | 0.306/0.284 (0.295) | 0.305/0.299 (**0.302**) | 0.290/0.281 (**0.285**) |
| Biology | Tempering | 0.266 | 0.252 | 0.248 | 0.216 |
| | Ensemble (LLM-SR) | 0.291/0.258 (0.274) | 0.287/0.241 (0.264) | 0.255/0.219 (0.237) | 0.221/0.176 (0.198) |
| | EvoDiverse | 0.322/0.293 (**0.307**) | 0.303/0.291 (**0.297**) | 0.277/0.269 (**0.272**) | 0.259/0.236 (**0.247**) |
| Chemistry | Tempering | 0.260 | 0.275 | 0.271 | 0.243 |
| | Ensemble (LLM-SR) | 0.272/0.265 (0.269) | 0.292/0.254 (0.270) | 0.286/0.232 (0.261) | 0.257/0.209 (0.233) |
| | EvoDiverse | 0.282/0.262 (**0.272**) | 0.299/0.266 (**0.283**) | 0.316/0.274 (**0.295**) | 0.277/0.259 (**0.268**) |
| Materials Science | Tempering | 0.216 | 0.208 | 0.201 | 0.188 |
| | Ensemble (LLM-SR) | 0.227/0.221 (0.224) | 0.227/0.220 (0.223) | 0.209/0.201 (0.205) | 0.191/0.176 (0.183) |
| | EvoDiverse | 0.232/0.227 (**0.229**) | 0.228/0.225 (**0.226**) | 0.221/0.213 (**0.217**) | 0.203/0.200 (**0.202**) |

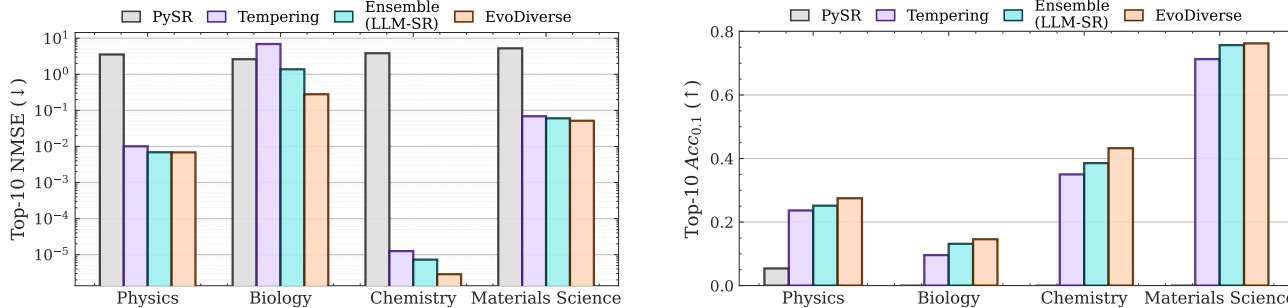

*Figure 9.* Performance comparison of EvoDiverse and LLM-based baselines against PySR across LLM-SRBench datasets (results averaged over backbones for LLM-based methods).

domains despite its strong performance as a non-LLM symbolic regressor, EvoDiverse shows substantial gains across all datasets.

Appendix Table 16 demonstrates a comprehensive quality comparison of EvoDiverse against LLM-based baselines (Tempering and Ensemble/LLM-SR) across all LLM-SRBench domains, evaluated separately on the `DeepSeek-V3.2` and `GPT-5` LLM backbones. We report both error-based metrics (Best NMSE and Top-10 NMSE, lower is better) and accuracy-based metrics based on threshold 0.1 (Best Acc0.1 and Top-10 Acc0.1, higher is better), averaged over problems per domain. Overall, these results in Table 2 and the detailed diversity results in Appendix Table 23 show that EvoDiverse not only improves best-case solution quality but also consistently raises the quality and diversity of the elite selected hypothesis set, meaning the exploration of better and higher-quality part of the hypothesis space which leads to outperforming prior LLM-based approaches across domains and backbone models.

### B.3.6. ABLATION: ENERGY FUNCTION CHOICE FOR PT SWAP

In equation discovery tasks, we instantiate the PT swap acceptance criterion using a log-transformed energy $h(x) = \log \text{MSE}(x)$, while the parent-selection Boltzmann weight is computed from the raw MSE, i.e. $p(x) \propto \exp(-\beta \text{MSE}(x))$. This choice is motivated by the wide dynamic range of MSE values encountered in symbolic regression (spanning $10^{-8}$ to $10^{+1}$ within a single run): raw differences $\text{MSE}(x_2) - \text{MSE}(x_1)$ are either dominated by outliers or vanish in low-error regimes, which prevents the adaptive scaling factor $\xi$ from maintaining a useful effective swap pressure across problems.

To verify this design choice, we re-ran EvoDiverse on the BIOPOPGROWTH (Biology) domain of LLM-SRBENCH with the swap criterion replaced by the raw-score variant $\log a = \xi(\beta_1 - \beta_2)(s_b - s_a)$, where $s = -\text{MSE}$, keeping all other components (selection, adaptive $\xi$, budget, and hyperparameters) identical. Since the log transform appears only in the

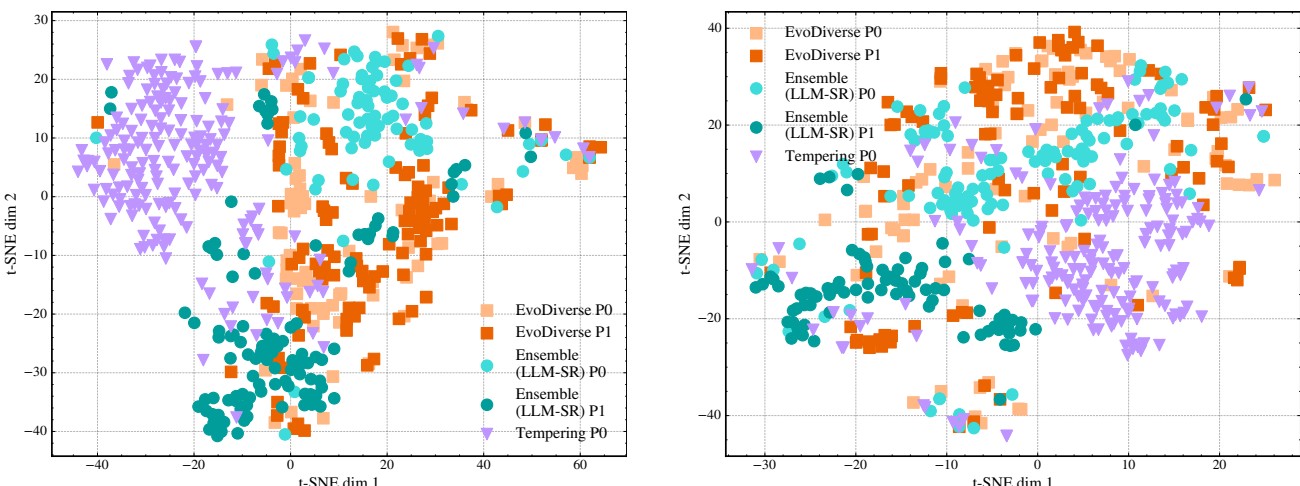

*Figure 10.* The qualitative embedding visualization of discovered equation program embeddings across different baselines on LLM-SRBench. Biology (**left**) and Chemistry (**right**) are shown as representative qualitative examples. EvoDiverse represents more diverse coverage of the embedding space

*Table 16.* Comparison of EvoDiverse with baselines on equation discovery across LLM-SRBench (Shojaee et al., 2025b) problems using DeepSeek-V3.2 and GPT-5 LLM backbones. Results are averaged per domain; **best values** per metric are bolded.

| Dataset | Method | *DeepSeek-V3.2* | | | | *GPT-5* | | | |
|---|---|---|---|---|---|---|---|---|---|
| | | Best NMSE ↓ | Top-10 NMSE ↓ | Best $Acc_{0.1}$↑ | Top-10 $Acc_{0.1}$↑ | Best NMSE ↓ | Top-10 NMSE ↓ | Best $Acc_{0.1}$↑ | Top-10 $Acc_{0.1}$↑ |
| Physics | Tempering | 0.0048 | 0.0062 | 0.291 | 0.230 | 0.0058 | 0.014 | 0.310 | 0.243 |
| | Ensemble (LLM-SR) | 0.0018 | 0.0061 | 0.333 | 0.240 | 0.0021 | **0.0077** | 0.233 | 0.263 |
| | EvoDiverse | **0.0015** | **0.0060** | **0.416** | **0.280** | **0.0017** | **0.0077** | **0.400** | **0.270** |
| Biology | Tempering | 2.601 | 13.41 | 0.083 | 0.079 | 0.402 | 0.403 | 0.125 | 0.113 |
| | Ensemble (LLM-SR) | 0.769 | 2.42 | 0.125 | **0.121** | 0.242 | 0.333 | 0.083 | 0.142 |
| | EvoDiverse | **0.136** | **0.307** | **0.250** | **0.121** | **0.103** | **0.253** | 0.174 | **0.171** |
| Chemistry | Tempering | $6.86e-6$ | $2.31e-5$ | 0.470 | 0.335 | $4.42e-8$ | $1.96e-6$ | 0.542 | 0.365 |
| | Ensemble (LLM-SR) | $3.34e-6$ | $1.44e-5$ | **0.588** | 0.306 | $5.69e-8$ | $1.61e-7$ | 0.588 | 0.465 |
| | EvoDiverse | **7.11e−7** | **5.65e−6** | **0.588** | **0.359** | **3.14e−8** | **8.62e−8** | **0.647** | **0.506** |
| Materials Science | Tempering | 0.0599 | 0.0897 | 0.688 | 0.641 | 0.0388 | 0.0476 | 0.700 | 0.785 |
| | Ensemble (LLM-SR) | 0.0479 | 0.0745 | 0.647 | 0.694 | 0.0384 | 0.0459 | 0.750 | 0.820 |
| | EvoDiverse | **0.0457** | **0.0578** | **0.706** | **0.700** | **0.0371** | **0.0451** | **0.900** | **0.825** |

swap step, this ablation only affects EvoDiverse; the Ensemble and Tempering baselines have no swap and are therefore unchanged, and we omit them from the ablation table.

**Discussion.** The ablation supports the design choice in the main paper. With the raw-score swap criterion, the cold pool only accepts swaps from the hot pool when the hot candidate is essentially as good as the current cold candidate (small $|s_b - s_a|$); otherwise the exponentially small acceptance probability dominates and the adaptive $\xi$ cannot recover a useful coupling across the temperature ladder. In contrast, the log-MSE swap operates on the *order of magnitude* of the error: a difference of one decade in MSE corresponds to a fixed contribution of $\xi(\beta_1 - \beta_2)\log 10$, regardless of whether both pools have $\mathrm{MSE}\approx 10^{-6}$ or both have $\mathrm{MSE}\approx 10^{-1}$. This preserves a meaningful inter-pool information transfer throughout the entire optimization trajectory, which matters in symbolic regression precisely because the population traverses several orders of magnitude in fitness during a single run.

We emphasise that this is a numerical-conditioning choice rather than a departure from the Parallel-Tempering template (Algorithms 1 and 3): the swap is still a Metropolis acceptance step with adaptive scaling that preferentially moves low-energy candidates from the hot pool to the cold pool. Importantly, the parent-selection mechanism remains $p(x) \propto \exp(-\beta\,\mathrm{MSE}(x))$ for *all three methods* (EvoDiverse, Ensemble, Tempering), so the inter-method comparison in the main paper is governed solely by whether (and how) cross-pool swaps occur; the additional log transform in the swap criterion is exclusive to EvoDiverse and does not affect the baselines.

*Table 17.* Ablation on the PT swap energy function for EvoDiverse on BIOPOPGROWTH (24 problems, DeepSeek-V3.2 backbone, $B=1000$ evaluations). **Log-MSE swap** corresponds to the main-paper formulation; **Raw-MSE swap** replaces the acceptance criterion with raw score differences $s_b - s_a$. Both variants share the same Boltzmann selection $p(x) \propto \exp(-\beta \, \mathrm{MSE}(x))$, so the comparison isolates the effect of the swap energy function. Best $\mathrm{Acc}_{0.1}$ follows the strict threshold from (Shojaee et al., 2025b): a problem is counted as "solved" if $\geq 95\%$ of test points lie within $10\%$ relative error of the ground truth.

| Swap variant for EvoDiverse | Best $\mathrm{Acc}_{0.1}$ (ID) $\uparrow$ |
|---|---|
| Raw-MSE: $\log a = \xi(\beta_1 - \beta_2)(s_b - s_a)$ | 0.042 |
| Log-MSE: $\log a = \xi(\beta_1 - \beta_2)\big(\log \mathrm{MSE}_a - \log \mathrm{MSE}_b\big)$ | **0.250** |

### B.3.7. ABLATION: METROPOLIS-HASTINGS SWAP

To isolate the role of the MH acceptance gate on equation discovery, we compare EvoDiverse against an Ensemble baseline (no swap) and a **Random Swap** variant that uses the identical two-pool architecture but accepts all proposed swaps unconditionally ($\xi=0$). Results on the Physics and Materials domains of LLM-SRBENCH using the DeepSeek-V3.2 backbone are shown in Appendix Table 18. Both swap-enabled methods improve over Ensemble, but the MH gate consistently delivers the strongest improvement in both diversity and accuracy. For example, on the Physics domain the Best $\mathrm{Acc}_{0.1}$ improves from 0.346 (Random Swap) to 0.416 (EvoDiverse; +20%).

*Table 18.* MH swap ablation on equation discovery. Results averaged over all datasets per domain and on the DeepSeek-V3.2 backbone.

| Domain | Method | Diversity $\uparrow$ | Best $\mathrm{Acc}_{0.1}$ $\uparrow$ | Top-10 $\mathrm{Acc}_{0.1}$ $\uparrow$ |
|---|---|---|---|---|
| Physics | Ensemble (no swap) | 0.289 | 0.333 | 0.240 |
| | Random Swap (no MH gate) | 0.295 | 0.346 | 0.263 |
| | EvoDiverse (MH swap) | **0.308** | **0.416** | **0.280** |
| Materials | Ensemble (no swap) | 0.224 | 0.688 | 0.641 |
| | Random Swap (no MH gate) | **0.229** | 0.684 | 0.675 |
| | EvoDiverse (MH swap) | **0.229** | **0.706** | **0.700** |

### B.3.8. HYPERPARAMETER SENSITIVITY

We study the sensitivity of EvoDiverse to the main PT hyperparameters on a subset of 10 Physics problems from LLM-SRBENCH using the DeepSeek-V3.2 backbone. Appendix Table 19 varies the cold-pool inverse temperature $\beta_1$ and the hot-pool inverse temperature $\beta_2$. Performance is stable for $\beta_1 \in \{0.8, 1.0\}$ paired with $\beta_2 \leq 0.1$; the method degrades only at the extremes ($\beta_1 = 1.5$ collapses pool diversity, while $\beta_2 = 0.5$ removes the temperature gap between the two pools). Appendix Table 20 varies the swap period $t_{\mathrm{swap}}$ and the per-swap batch size $k$. A swap period of 2–5 works well; longer periods reduce diversity due to infrequent exchange. Batch sizes $k \geq 3$ are robust, with diminishing returns beyond $k = 5$. Across all settings the behaviour is graceful degradation, not catastrophic failure, supporting the use of the defaults marked with [†] across all three domains without task-specific tuning.

*Table 19.* Sensitivity to selection temperatures on equation discovery (10 Physics problems, using DeepSeek-V3.2). The default setting is marked with [†].

| $\beta_1$ (cold) | $\beta_2$ (hot) | Diversity $\uparrow$ | Best $\mathrm{Acc}_{0.1}$ $\uparrow$ | Top-10 $\mathrm{Acc}_{0.1}$ $\uparrow$ |
|---|---|---|---|---|
| 0.5 | 0.01 | 0.315 | 0.3 | 0.1 |
| 0.8[†] | 0.01[†] | 0.302 | 0.5 | 0.3 |
| 1.0 | 0.01 | 0.289 | 0.5 | 0.3 |
| 1.5 | 0.01 | 0.261 | 0.4 | 0.1 |
| 0.8 | 0.05 | 0.302 | 0.5 | 0.3 |
| 0.8 | 0.1 | 0.302 | 0.4 | 0.3 |
| 0.8 | 0.2 | 0.285 | 0.5 | 0.2 |
| 0.8 | 0.5 | 0.283 | 0.3 | 0.1 |

*Table 20.* Sensitivity to swap period and batch size $k$ on equation discovery (10 Physics problems, using DeepSeek-V3.2). The default setting is marked with [†].

| Swap Period | Swap Size $k$ | Diversity ↑ | Best Acc$_{0.1}$ ↑ | Top-10 Acc$_{0.1}$ ↑ |
|---|---|---|---|---|
| 2 | 5 | 0.321 | 0.5 | 0.3 |
| 5[†] | 5[†] | 0.302 | 0.5 | 0.3 |
| 10 | 5 | 0.295 | 0.3 | 0.2 |
| 20 | 5 | 0.284 | 0.3 | 0.2 |
| 5 | 1 | 0.289 | 0.3 | 0.3 |
| 5 | 3 | 0.291 | 0.4 | 0.3 |
| 5[†] | 5[†] | 0.302 | 0.5 | 0.3 |
| 5 | 10 | 0.302 | 0.5 | 0.3 |

### B.3.9. GENERALIZATION TO MORE TEMPERATURE LEVELS

The PT framework is not restricted to two temperature pools. To verify that the gains carry over to deeper temperature ladders, we replicate the equation-discovery setup with three pools at $\beta \in \{0.8, 0.2, 0.01\}$ on the Physics subset of LLM-SRBENCH using the DeepSeek-V3.2 backbone. The three-temperature variant marginally improves both diversity and accuracy over the two-temperature default (Appendix Table 21), confirming that the architecture generalizes to more pools. We use two pools by default because it offers the simplest setup with a favourable cost–benefit tradeoff under a fixed total oracle budget.

*Table 21.* Effect of number of temperature levels on equation discovery (Physics subset). "2-Temp" uses $\beta \in \{0.8, 0.01\}$ (the default); "3-Temp" adds an intermediate level $\beta \in \{0.8, 0.2, 0.01\}$. Experiments conducted on DeepSeek-V3.2.

| Method | Diversity ↑ | Best Acc$_{0.1}$ ↑ | Top-10 Acc$_{0.1}$ ↑ |
|---|---|---|---|
| Ensemble (2 pools) | 0.289 | 0.333 | 0.240 |
| EvoDiverse (2-Temp) | 0.308 | 0.416 | 0.280 |
| EvoDiverse (3-Temp) | 0.316 | 0.421 | 0.280 |

### B.3.10. STATISTICAL SIGNIFICANCE

Appendix Table 22 reports paired one-sided $t$-tests of EvoDiverse over the Tempering and Ensemble baselines on equation discovery. Both improvements are highly significant ($p < 0.001$).

*Table 22.* Statistical significance of EvoDiverse over baselines on equation discovery (paired $t$-test over seeds/datasets, one-sided). Results averaged over both LLM backbones.

| Comparison | Best Acc$_{0.1}$ $p$-value | Top-10 Acc$_{0.1}$ $p$-value |
|---|---|---|
| vs. Tempering | $< 0.001$ | $< 0.001$ |
| vs. Ensemble (LLM-SR) | $< 0.001$ | $< 0.001$ |

### B.3.11. DIAGNOSTIC CURVES

Appendix Figure 11 plots the adaptive powering factor $\xi$ and the empirical swap acceptance rate over swap iterations for EvoDiverse on the four LLM-SRBENCH domains and both LLM backbones. The acceptance rate stabilises rapidly near the target ($\sim 30\%$), while $\xi$ increases monotonically as the underlying score distributions sharpen during search. The qualitative behaviour matches that observed for molecular and algorithm discovery (Appendix Figure 8 and 14).

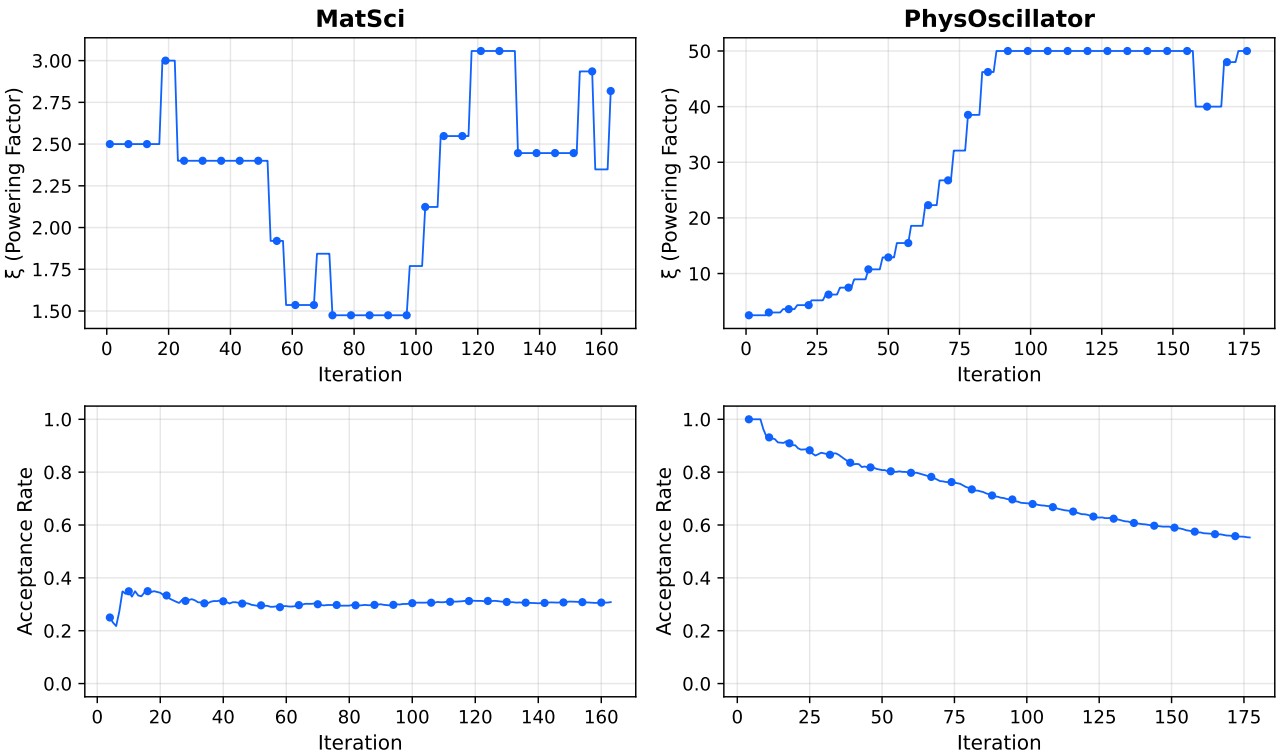

*Figure 11.* Diagnostic curves for EvoDiverse on equation discovery.

## B.4. Algorithm Discovery

### B.4.1. TASK SPECIFICATION AND IMPLEMENTATION DETAILS

The circle packing task requires placing $n = 26$ circles within a unit square to maximize the sum of radii while ensuring no circles overlap and all remain fully within the boundary (Appendix Figure 12). The search space consists of Python programs that construct circle configurations (centers and radii). This follows the standard formulation from prior work (Novikov et al., 2025; Sharma, 2025). We use the verification script from OpenEvolve (Sharma, 2025), which allows $10^{-6}$ numerical slack for floating-point precision. Programs are executed with 30-second timeout limits and validated against geometric constraints; invalid configurations (overlapping circles or boundary violations) receive a penalty score of 0.

All methods use DeepSeek-V3.2 as the LLM backend with identical prompting strategies. Programs are sampled from a database of previous solutions using fitness-proportional selection, with 1,000 total evaluations distributed across all methods.

**Score function.** Each program $x$ is executed and validated against the geometric constraints; we then define its score as

$$\ell(x) := \frac{\sum_{i=1}^{n} r_i(x)}{R^\star} \cdot \mathbf{1}[x \text{ valid}] \ \in \ [0, 1], \tag{26}$$

where $r_i(x)$ are the radii produced by program $x$, $R^\star = 2.635$ is the reference upper bound on the sum of radii for $n = 26$ circles in a unit square used for normalization, and $\mathbf{1}[x \text{ valid}] \in \{0, 1\}$ indicates that all geometric constraints are satisfied within the $10^{-6}$ tolerance. Larger $\ell$ corresponds to better programs. The corresponding energy used in the PT swap criterion is $h(x) := -\ell(x)$.

**EvoDiverse Configuration.** Following the cross-task convention in Section B.1, both mating-pool sampling and the PT swap criterion are expressed in terms of $\ell(\cdot)$ (equivalently $h = -\ell$). Different from equation discovery, the algorithm-discovery score $\ell$ is bounded in $[0, 1]$ and does not span orders of magnitude, so we do not apply the log-MSE energy adjustment introduced in Section B.3.6 and keep the standard PT swap criterion. Other key design choices are:

- **Pool temperatures**: Cold pool ($\beta = 1.0$, $T = 1.0$), Hot pool ($\beta = 0.25$, $T = 4.0$)
- **Swap mechanism**: Unit swaps every 5 iterations with Metropolis-Hastings acceptance

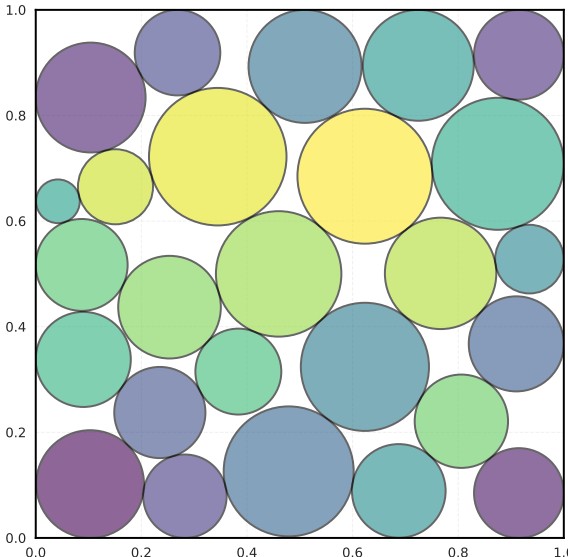

*Figure 12.* Circle packing problem illustration showing the best solution discovered by EvoDiverse ($n = 26$, sum of radii = 2.5461).

- **Adaptive energy scaling**: Initial $\xi = 2.5$, with target swap rate $0.3 \pm 0.1$
- **Selection**: Mating-pool sampling uses the pool-specific Boltzmann rule $p_i^{\text{mating}}(x) \propto \exp(\ell(x)/T_i)$ defined in Section B.1; next-generation update is deterministic (append the child and remove the worst-scoring candidate once the pool capacity of 500 programs per pool is exceeded).
- **LLM generation**: Cold pool uses generation temperature 0.7 (top-p 0.8), Hot pool uses 1.0 (top-p 0.95)

**Baseline Configurations.** All baselines use the same total evaluation budget (1000 programs) and identical LLM prompting:

- **EA**: Standard evolutionary algorithm with one population of 1000 programs; mating-pool sampling uses $p^{\text{mating}}(x) \propto \exp(\ell(x)/T)$ with $T = 1$ (single pool, no inter-pool temperature differentiation).
- **Island**: Two pools of 500 programs each, both at $T = 1$, with ring migration every 5 iterations; each pool independently uses the same mating-pool Boltzmann sampling. This represents traditional island-model evolutionary algorithms with uncontrolled coupling.
- **Ensemble**: Two isolated pools of 500 programs each, both at $T = 1$, with no migration (migration interval set to $10^6$); each pool independently uses the same mating-pool Boltzmann sampling. This represents an ensemble of independent runs without information exchange.

### B.4.2. DETAILED DIVERSITY METRICS

Table 23 provides comprehensive diversity analysis including both lexical (TF-IDF) and semantic (CodeBERT (Feng, 2020)) measures, per-pool breakdowns, and pool contribution to the elite set. The results confirm the patterns discussed in the main text: EvoDiverse achieves productive pool differentiation with asymmetric elite contributions (Cold pool: 68%), Island EA homogenizes pools with symmetric contributions (50-50 split), and Ensemble EA exhibits catastrophic asymmetry (one pool contributes 0%).

*Table 23.* Comprehensive diversity metrics for circle packing. Per-pool format: Pool 0 / Pool 1 (Cold / Hot for EvoDiverse).

| Method | Lexical (TF-IDF) | | Semantic (CodeBERT) | | Pool Origin (Top-100) |
|---|---|---|---|---|---|
| | Overall | Per-Pool | Overall | Per-Pool | |
| EvoDiverse | **0.78** | 0.54 / 0.77 | 1.10 | 0.76 / 1.22 | 68% / 32% |
| EA | 0.61 | — | 1.20 | — | — |
| Island | 0.48 | 0.44 / 0.48 | 0.73 | 0.74 / 0.71 | 50% / 50% |
| Ensemble | 0.76 | 0.51 / 0.50 | 1.27 | 0.42 / 0.99 | 0% / 100% |

### B.4.3. DIVERSITY LANDSCAPE VISUALIZATION

Appendix Figure 13 shows PCA projections of the final populations for all methods using TF-IDF embeddings. The visualization consists of two panels: the left panel colors points by pool identity (Cold/Hot for EvoDiverse , Pool 0/Pool 1 for baselines), while the right panel indicates program scores, revealing the relationship between diversity and optimization quality.

The visualizations reveal distinct population structures across methods. **EvoDiverse** shows two partially overlapping clusters: the Cold pool forms clusters in high-score regions, while the Hot pool explores a broader area of the solution space. The overlap region represents solutions that successfully transferred between pools via thermodynamic swaps. **Island EA** displays a single merged cluster with extensive color mixing, indicating that frequent migration synchronized both pools into an identical population. Both Pool 0 and Pool 1 occupy the same region with similar score distributions, confirming the homogenization effect observed in the diversity metrics. **Ensemble EA** exhibits two disconnected clusters with no overlap: Pool 0 appears as a tight, isolated cluster in a low-score region, while Pool 1 shows normal evolutionary spread across medium-to-high score regions.

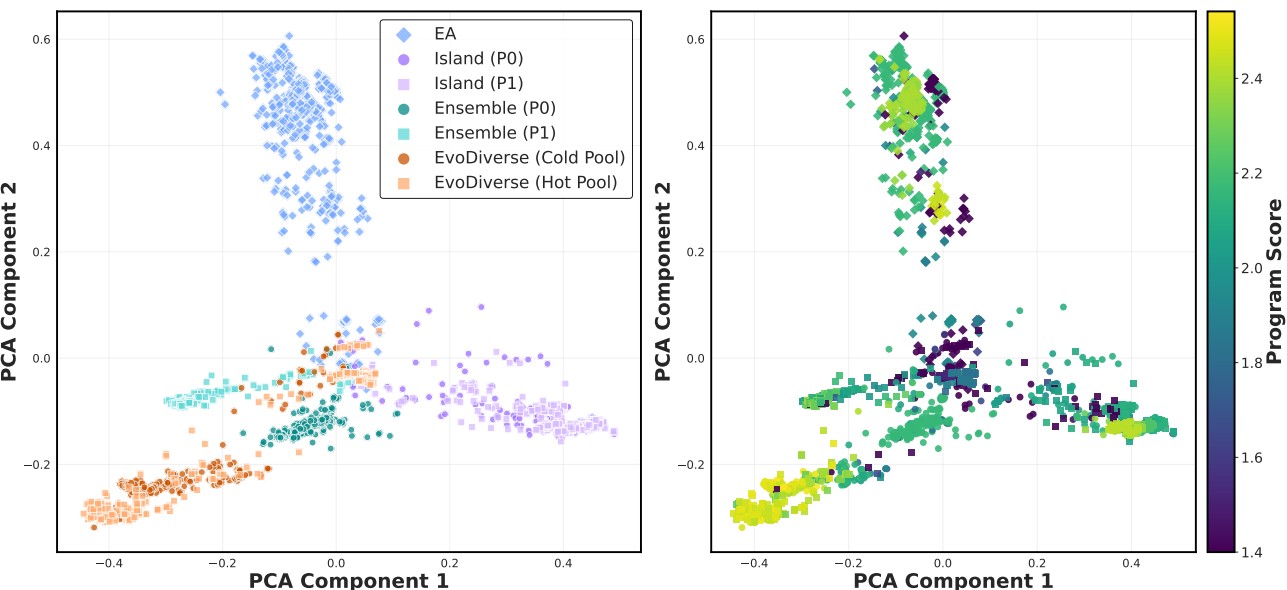

*Figure 13.* PCA visualization of final populations using TF-IDF embeddings. **Left:** Points colored by method and pool: EvoDiverse shows differentiated Cold and Hot pools with partial overlap; Island shows homogenized pools; Ensemble shows disconnected pools. **Right:** Same projection colored by program scores.

### B.4.4. ABLATION: METROPOLIS-HASTINGS SWAP

To isolate the contribution of the MH acceptance gate in the PT swap, we compare EvoDiverse against two natural variants of the same two-pool architecture: **Ensemble** (no swap; two independent pools), and **Random Swap** (the identical EvoDiverse architecture with $T = [1, 4]$ but with the MH gate disabled, all proposed swaps are accepted unconditionally, equivalent to $\xi = 0$). As shown in Appendix Table 24, both Random Swap and EvoDiverse improve over the Ensemble baseline that lacks inter-pool communication, but Random Swap collapses the temperature differentiation between the two pools: its hot-pool diversity (0.591) and cold-pool diversity (0.559) are nearly equal, because indiscriminate exchange floods the cold pool with hot-pool exploratory programs. The MH gate preserves the cold/hot asymmetry (EvoDiverse's hot pool diversity 0.712 vs. cold pool 0.529) and yields both the best optimization quality and the highest overall diversity. The diversity reported here is recomputed over all unique programs in the evolution trace (for consistency with the Random Swap variant), which yields slightly lower values than the MAP-Elites–archive–based diversity reported in Table 3 (e.g., 0.714 vs. 0.78 for EvoDiverse), but the relative ranking across methods is identical.

*Table 24.* MH swap ablation on circle packing ($n = 26$). "Random Swap" uses the identical EvoDiverse architecture ($T = [1, 4]$) but replaces the MH gate with an unconditional swap ($\xi=0$, always accept). Diversity values here are computed over all unique programs in the evolution trace and are therefore slightly lower than the MAP-Elites–archive values in the main paper, while the ranking is preserved.

| Method | Best Sum ↑ | Top-100 Avg ↑ | TF-IDF Diversity ↑ |
|---|---|---|---|
| EA (single pool) | 2.4986 | 2.4302 | 0.596 |
| Island EA (frequent migration) | 2.4247 | 2.4241 | 0.471 |
| Ensemble EA (no swap) | 2.4105 | 2.3330 | 0.691 |
| Random Swap (no MH gate) | 2.4507 | 2.4507 | 0.576 |
| **EvoDiverse** (MH swap) | **2.5461** | **2.5138** | **0.714** |

### B.4.5. COMPUTATIONAL COST

All methods on the circle-packing task share the same oracle budget of 1,000 program evaluations and therefore make the same number of LLM calls. Appendix Table 25 reports the empirical token usage, median per-iteration time, estimated wall-clock time, and invalid-proposal rate. Total token usage is comparable across methods (11–14 M), and median iteration times only vary modestly (20–39 s); the variation reflects program length differences across runs and server load rather than method-level overhead. EvoDiverse exhibits the lowest invalid-proposal rate (0.4%), supporting the interpretation that the MH gate selectively transfers higher-quality candidates and prevents borderline-invalid programs from propagating between pools.

*Table 25.* Computational cost on circle packing ($n = 26$). Oracle budget: 1,000 program evaluations.

| Method | LLM Calls | Est. Total Tokens (M) | Median Iter Time (s) | Est. Wall-Clock (h) | Invalid Rate (%) | Best Sum ↑ |
|---|---|---|---|---|---|---|
| EA (1 pool) | 1,000 | 12.6 | 39 | ~11 | 8.8 | 2.4986 |
| Island EA | 1,000 | 13.7 | 22 | ~6 | 2.8 | 2.4247 |
| Ensemble EA | 1,000 | 12.0 | 27 | ~7 | 0.9 | 2.4105 |
| Random Swap | 1,000 | 11.0 | 20 | ~5 | 3.6 | 2.4507 |
| **EvoDiverse** | 1,000 | 12.9 | 36 | ~10 | 0.4 | **2.5461** |

### B.4.6. DIAGNOSTIC CURVES

Appendix Figure 14 plots the adaptive powering factor $\xi$ and the empirical swap acceptance rate over swap iterations for EvoDiverse on circle packing. $\xi$ adapts monotonically to maintain the acceptance rate close to the target (30–40%), consistent with the discussion in Section B.4.1 and with the analogous diagnostics in molecular and equation discovery (Appendix Figure 8 and 11).

### B.4.7. T-SNE VISUALIZATION

Appendix Figure 15 provides a t-SNE projection of the TF-IDF embeddings of all programs generated by each method on the circle-packing task, complementing the PCA projection in the main paper. The qualitative patterns visible under PCA persist under t-SNE: EvoDiverse exhibits differentiated but overlapping cold and hot pool clusters, the Island baseline collapses into a single homogenized cluster, and Ensemble splits into disconnected clusters that fail to share information.

### B.4.8. DISCOVERED SOLUTION

Appendix Figure 12 visualizes the best circle packing configuration discovered by EvoDiverse with a sum of radii of 2.5461. The solution exhibits structured geometric organization: 4 circles positioned at corners with optimized offset from boundaries, 12 circles distributed along edges (3 per edge between corners), and 10 circles arranged in the interior region. The configuration demonstrates sophisticated space utilization with circles of varying sizes packed efficiently while satisfying all constraints.

Program 1 shows the discovered code. The algorithm initializes circles using a geometric constructor with optimized placement parameters, then applies multi-scale local search with progressively finer step sizes combined with random

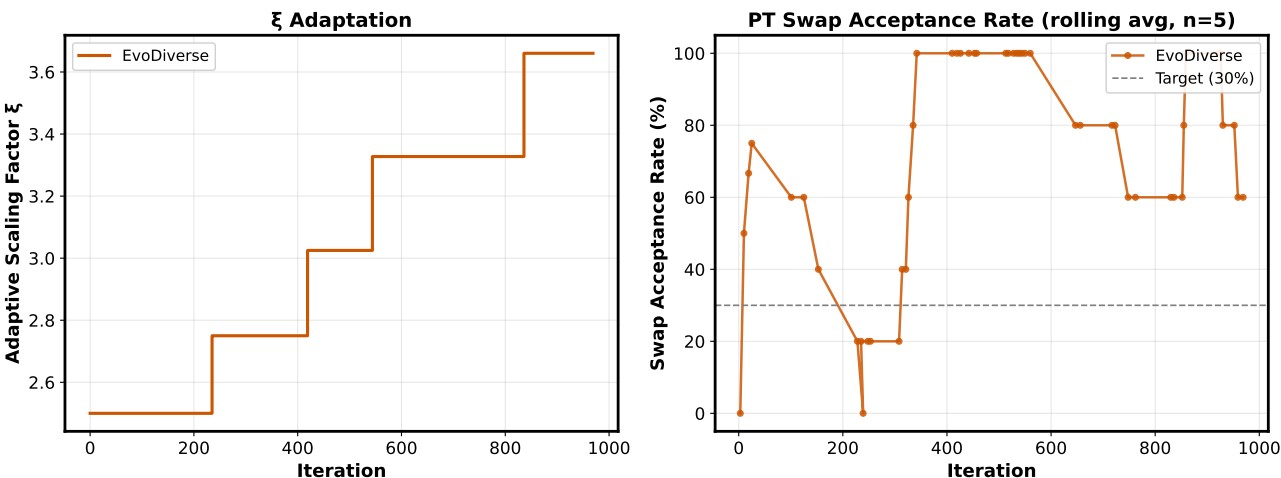

*Figure 14.* Diagnostic curves for EvoDiverse on circle packing. The adaptive powering factor $\xi$ (left) and the swap acceptance rate (right) over swap iterations.

position swaps to escape local optima.

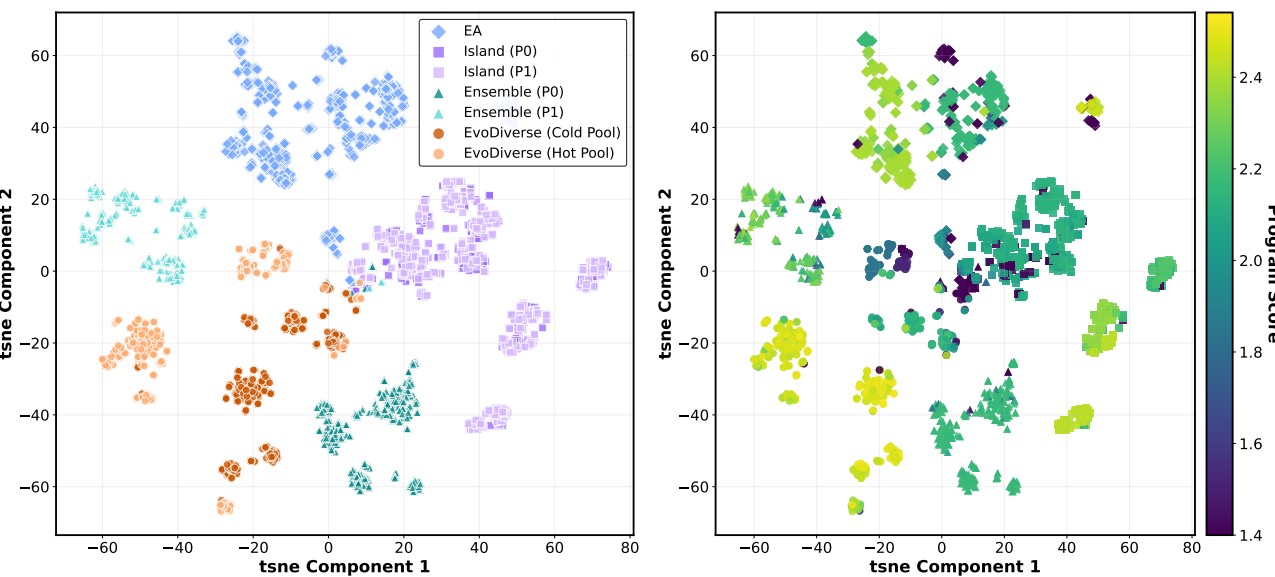

*Figure 15.* t-SNE visualization of final populations on circle packing using TF-IDF embeddings, complementing the PCA projection in the main paper.

```python
import numpy as np

def construct_packing():
    n = 26
    centers = np.zeros((n, 2))
    idx = 0

    # Place 4 circles at corners with optimized offset (from top performer Program 1)
    offset = 0.065
    corners = [(offset, offset), (1-offset, offset), (offset, 1-offset), (1-offset, 1-offset)]
    for i in range(4):
        centers[idx] = corners[i]
        idx += 1

    # Place 3 additional circles per edge (total 4 per edge including corners)
    for i in range(1, 4):
        t = i / 4.0
        x = offset + t * (1 - 2*offset)
        centers[idx] = (x, offset); idx += 1
        centers[idx] = (x, 1-offset); idx += 1
        y = offset + t * (1 - 2*offset)
        centers[idx] = (offset, y); idx += 1
        centers[idx] = (1-offset, y); idx += 1

    # Place remaining 10 circles in interior using 3 rows: 4,3,3
    inner_margin = 0.13  # Adjusted from top performer
    row_counts = [4, 3, 3]
    total_rows = 3
    spacing = (1 - 2*inner_margin) / 3
    for row in range(total_rows):
        y = inner_margin + (row + 1) * (1 - 2*inner_margin) / (total_rows + 1)
        if row % 2 == 0:
            start_x = inner_margin
        else:
            start_x = inner_margin + spacing / 2.0
        for col in range(row_counts[row]):
            x = start_x + col * spacing
            if idx < n:
                centers[idx] = (x, y)
                idx += 1
    if idx < n:
        centers[idx:] = [0.5, 0.5]
    centers = np.clip(centers, 0.001, 0.999)

    # Compute radii
    radii = compute_max_radii(centers)
```

```
48        # Enhanced local improvement with adaptive step sizes (from top performer)
49        best_c = centers.copy()
50        best_r = radii.copy()
51        best_sum = np.sum(radii)
52        steps = [0.04, 0.02, 0.01, 0.005]
53        for step in steps:
54            improved = True
55            while improved:
56                improved = False
57                order = np.random.permutation(n)
58                for i in order:
59                    # Try moves in a 5x5 grid for larger steps, 3x3 for smaller
60                    num_points = 5 if step >= 0.02 else 3
61                    for dx in np.linspace(-step, step, num_points):
62                        for dy in np.linspace(-step, step, num_points):
63                            if dx == 0 and dy == 0: continue
64                            new = best_c[i] + [dx, dy]
65                            new = np.clip(new, 0.001, 0.999)
66                            temp = best_c.copy()
67                            temp[i] = new
68                            r_temp = compute_max_radii(temp)
69                            s = np.sum(r_temp)
70                            if s > best_sum:
71                                best_c, best_r, best_sum = temp, r_temp, s
72                                improved = True
73            # Random swaps with more attempts
74            for _ in range(8):
75                i, j = np.random.choice(n, 2, replace=False)
76                temp = best_c.copy()
77                temp[i], temp[j] = best_c[j], best_c[i]
78                r_temp = compute_max_radii(temp)
79                s = np.sum(r_temp)
80                if s > best_sum:
81                    best_c, best_r, best_sum = temp, r_temp, s
82
83        # Final fine-tuning pass with even smaller steps
84        best_r = compute_max_radii(best_c)
85        best_sum = np.sum(best_r)
86        for step in [0.002, 0.001]:
87            order = np.random.permutation(n)
88            for i in order:
89                for dx in np.linspace(-step, step, 3):
90                    for dy in np.linspace(-step, step, 3):
91                        if dx == 0 and dy == 0: continue
92                        new = best_c[i] + [dx, dy]
93                        new = np.clip(new, 0.001, 0.999)
94                        temp = best_c.copy()
95                        temp[i] = new
96                        r_temp = compute_max_radii(temp)
97                        s = np.sum(r_temp)
98                        if s > best_sum:
99                            best_c, best_r, best_sum = temp, r_temp, s
100       return best_c, best_r, best_sum
101
102   def compute_max_radii(centers):
103       n = centers.shape[0]
104       border_dists = np.minimum(centers, 1 - centers)
105       radii = np.min(border_dists, axis=1)
106       # Precompute distance matrix
107       diff = centers[:, np.newaxis, :] - centers[np.newaxis, :, :]
108       dist = np.sqrt(np.sum(diff**2, axis=2))
109       np.fill_diagonal(dist, np.inf)
110       # Iterative scaling
111       for _ in range(100):
112           pair_sum = radii[:, None] + radii
113           if not np.any(pair_sum > dist):
114               break
115           scale = np.ones(n)
116           for i in range(n):
117               for j in range(i+1, n):
118                   if radii[i] + radii[j] > dist[i, j]:
119                       s = dist[i, j] / (radii[i] + radii[j])
120                       if s < scale[i]:
121                           scale[i] = s
122                       if s < scale[j]:
123                           scale[j] = s
124           radii *= scale
125           radii = np.minimum(radii, np.min(border_dists, axis=1))
126       # Greedy expansion
127       for i in range(n):
128           max_r = np.min(border_dists[i])
```

```
129        for j in range(n):
130            if i == j: continue
131            max_r = min(max_r, dist[i, j] - radii[j])
132        radii[i] = max_r
133    # Final safety
134    for _ in range(10):
135        for i in range(n):
136            for j in range(i+1, n):
137                if radii[i] + radii[j] > dist[i, j]:
138                    s = dist[i, j] / (radii[i] + radii[j])
139                    radii[i] *= s
140                    radii[j] *= s
141        radii = np.minimum(radii, np.min(border_dists, axis=1))
142    return radii
```

*Program 1.* Best circle packing program discovered by EvoDiverse. The program uses a constructor-based approach with optimized geometric placement.

# C. Prompts

We show the prompts of EvoDiverse in three tasks.

---

**Molecular Discovery**

**hot_pool**
I have two molecules and their JNK3 scores (higher is better).  The JNK3 score measures the inhibitory ability of a molecule against c-Jun N-terminal kinase 3 (JNK3).

(Smiles of Parent A, objective score of Parent A) (Smiles of Parent B, objective score of Parent B)

Requirements:
- Exploration:  Propose a molecule that balances novelty (chemical space exploration) and competitiveness (likely high/maintained score).
- In your response, please provide a rationale summarizing:  (i) which space is being explored, (ii) why score should remain competitive.  and then propose the molecule.  You need to conclude your answer with the sentence below (replacing the placeholder with the SMILES of your proposed molecule):

Based on the above analysis, the proposed molecule is:  <box>Molecule SMILES</box>.

**cold_pool**
I have two molecules and their JNK3 scores (higher is better).  The JNK3 score measures the inhibitory ability of a molecule against c-Jun N-terminal kinase 3 (JNK3).

(Smiles of Parent A, objective score of Parent A) (Smiles of Parent B, objective score of Parent B)

Requirements:
- Please propose a molecule that has a higher target score based on your knowledge.  You can either make crossover and mutations based on the given molecules or just propose a new molecule based on your knowledge.
- In your response, please provide a concise rationale summarizing how to achieve a higher target score and then propose the molecule.  You need to conclude your answer with the sentence below (replacing the placeholder with the SMILES of your proposed molecule):

Based on the above analysis, the proposed molecule is:  <box>Molecule SMILES</box>.

---

In the equation discovery, we use the same prompt originally used in LLM-SR framework (Shojaee et al., 2025a). Below two examples of this prompt style for multi-variable equations of force and energy are provided.

---

**Equation Discovery**

Find the mathematical function skeleton that represents force, given data on mass, and acceleration.

(Program v0, score:  -0.0456)
@equation.evolve
def equation(mass:  np.ndarray, acceleration:  np.ndarray, params:  np.ndarray) -> np.ndarray:
""" Mathematical function for force
Args:
mass:  A numpy array representing observations of mass.
acceleration:  A numpy array representing observations of acceleration.
params:  Array of numeric constants or parameters to be optimized
Return:
A numpy array representing force as the result of applying the mathematical function to the inputs.
"""

---

```
force = params[0] * mass + params[1] * acceleration + params[2]
return force

(Program v1, score: -0.0321)
@equation.evolve
def equation_v1(mass: np.ndarray, acceleration: np.ndarray, params: np.ndarray) -> np.ndarray:
""" Improved version of 'equation_v0'.
"""
force = params[0] * mass * acceleration + params[1] * np.sin(mass) + params[2]
return force

@equation.evolve
def equation_v2(mass: np.ndarray, acceleration: np.ndarray, params: np.ndarray) -> np.ndarray:
""" Improved version of 'equation_v1'.
"""
```

## Equation Discovery

```
Find the mathematical function skeleton that represents energy, given data on velocity, mass, and height.

(Program v0, score: -0.0789)
@equation.evolve
def equation(velocity: np.ndarray, mass: np.ndarray, height: np.ndarray, params: np.ndarray) ->
np.ndarray:
""" Mathematical function for energy
Args:
velocity: A numpy array representing observations of velocity.
mass: A numpy array representing observations of mass.
height: A numpy array representing observations of height.
params: Array of numeric constants or parameters to be optimized
Return:
A numpy array representing energy as the result of applying the mathematical function to the inputs.
"""
energy = params[0] * velocity + params[1] * mass + params[2] * height + params[3]
return energy

(Program v1, score: -0.0567)
@equation.evolve
def equation_v1(velocity: np.ndarray, mass: np.ndarray, height: np.ndarray, params: np.ndarray) ->
np.ndarray:
""" Improved version of 'equation_v0'.
"""
energy = params[0] * mass * velocity**2 + params[1] * mass * height + params[2]
return energy

@equation.evolve
def equation_v2(velocity: np.ndarray, mass: np.ndarray, height: np.ndarray, params: np.ndarray) ->
np.ndarray:
""" Improved version of 'equation_v1'.
"""
```

We adopt the OpenEvolve (Sharma, 2025) prompting for algorithm discovery, which consists of several prompts including a system message and task-specific prompts that can operate in two modes: full rewrite or targeted diff-based improvements as shown below.

## Algorithm Discovery

```
System Message
You are an expert software developer tasked with iteratively improving a codebase.
Your goal is to maximize the FITNESS SCORE while exploring diverse solutions across feature dimensions.
The system maintains a collection of diverse programs – both high fitness AND diversity are valuable.
Full Rewrite Prompt
# Current Program Information
- Fitness: {fitness_score}
- Feature coordinates: {feature_coords}
- Focus areas: {improvement_areas}

{artifacts}
```

```
# Program Evolution History
{evolution_history}

# Current Program
``````{language}
{current_program}
`````
# Task
Rewrite the program to improve its FITNESS SCORE.
The system maintains diversity across these dimensions: {feature_dimensions}
Different solutions with similar fitness but different features are valuable.
Provide the complete new program code.

IMPORTANT: Make sure your rewritten program maintains the same inputs and outputs
as the original program, but with improved internal implementation.

````{language}
# Your rewritten program here
````
Diff-Based Improvement Prompt
# Current Program Information
- Fitness: {fitness_score}
- Feature coordinates: {feature_coords}
- Focus areas: {improvement_areas}

{artifacts}
# Program Evolution History
{evolution_history}

# Current Program
````{language}
{current_program}
```
# Task
Suggest improvements to the program that will improve its FITNESS SCORE.
The system maintains diversity across these dimensions: {feature_dimensions}
Different solutions with similar fitness but different features are valuable.

You MUST use the exact SEARCH/REPLACE diff format shown below to indicate changes:
<<<<<<< SEARCH
# Original code to find and replace (must match exactly)
=======
# New replacement code
>>>>>>> REPLACE

Example of valid diff format:
<<<<<<< SEARCH
for i in range(m):
for j in range(p):
for k in range(n):
C[i, j] += A[i, k] * B[k, j]
=======
# Reorder loops for better memory access pattern
for i in range(m):
for k in range(n):
for j in range(p):
C[i, j] += A[i, k] * B[k, j]
>>>>>>> REPLACE

You can suggest multiple changes. Each SEARCH section must exactly match code in the current program.
Be thoughtful about your changes and explain your reasoning thoroughly.

IMPORTANT: Do not rewrite the entire program - focus on targeted improvements.
```