# OpenReview forum: "Towards Diverse Scientific Hypothesis Search with Large Language Models"
_ICML.cc/2026/Conference — ICML 2026 regular_

### Official Review · Reviewer_ocBA · 2026-03-04

**Soundness:** 2
**Presentation:** 3
**Significance:** 2
**Originality:** 3
**Overall Recommendation:** 5
**Confidence:** 3

**Summary:**

The authors investigate a major question regarding the lack of diversity in LLM-based evolutionary search for scientific hypotheses. The research's major contribution is the proposal of EvoDiverse, a framework inspired by parallel tempering that maintains multiple populations at different temperatures and exchanges candidates using a Metropolis-Hastings criterion. The method is evaluated on molecular discovery, equation discovery, and algorithm discovery tasks, demonstrating improvements in solution quality and diversity.

**Compliance With Llm Reviewing Policy:**

Affirmed.

**Final Justification:**

I acknowledge the authors' thorough rebuttal. The newly provided ablation studies on ξ sensitivity (Tables 5–6), the same-prompt ablation (Table 3), the per-generation budget breakdown (Table 7), and the invalid proposal rate analysis effectively address my main technical concerns (Q1, Q3, Q5, Q6). The three-temperature experiment (Table 2) provides a reasonable empirical justification for the two-pool design (Q2), though a brief theoretical discussion in the revised paper would further strengthen the narrative.

Regarding functional diversity (Q4), the evidence from unseen-objective filtering in molecular discovery and the qualitatively distinct strategies in algorithm discovery is encouraging, though not fully conclusive. I appreciate the authors' candor in noting this as future work.

Overall, the rebuttal has substantively strengthened the paper. I am adjusting my score upward accordingly.

**Key Questions For Authors:**

1. How sensitive is the optimization trajectory and final diversity to the initialization of $\xi$ and the choice of target swap rate?

2. What is the empirical or theoretical justification for limiting the experiments to only two temperature pools instead of a broader temperature ladder?

3. Are the per-generation mating pool sizes and offspring generation budgets strictly identical between EvoDiverse and the multi-population baselines?

4. Does the parallel tempering mechanism promote true functional diversity, or does it merely generate structurally distinct variants mapping to the same behavior?

5. How much of the performance gain in molecular discovery is attributable to the different LLM prompts versus the Metropolis-Hastings swapping mechanism?

6. What is the rejection rate of invalid proposals in the hot pool compared to the cold pool, and how does this affect the overall wall-clock time?

**Limitations:**

The authors briefly mention the need for real-world experimental validation in the appendix but lack a dedicated limitations section in the main text. They largely dismiss potential societal impacts and do not discuss the computational overhead or prompt sensitivity of their framework. Suggestion: Add a dedicated "Limitations" section discussing the computational cost of maintaining multiple LLM-driven populations and the sensitivity of the method to initial prompt engineering.

**Strengths And Weaknesses:**

**Strengths**

1. The conceptual mapping of evolutionary algorithm selection pressure to thermodynamic temperature is theoretically sound. It provides a principled mechanism to balance exploration and exploitation in LLM-driven search.

2. The empirical validation spans three distinct scientific domains with different representation spaces. This demonstrates the generalizability of the proposed framework across SMILES, mathematical expressions, and Python code.

**Weaknesses**

1. The adaptive mechanism for the scaling factor $\xi$ relies on a fixed window size and target swap rate. The paper lacks an ablation study on how sensitive the final performance is to these initial hyperparameters.

**Suggestion:** Add an ablation study varying the window size, target rate, and the initial $\xi$ value.

2. The methodology section claims the approach generalizes to multiple temperature levels, but experiments only utilize two pools. It is unclear if adding more intermediate temperatures provides marginal benefits or just decreases sample efficiency.

**Suggestion:** Include an experiment or theoretical discussion analyzing the trade-off of using three or more temperature pools.

3. In the multi-population baselines, it is not entirely clear if the effective mating pool size per generation is strictly controlled to match EvoDiverse. Unequal mating pool dynamics could skew the diversity metrics and convergence speed.

**Suggestion:** Explicitly detail the per-generation parent and offspring counts for all baselines in the main text.

4. The diversity metrics are evaluated post-hoc but are not explicitly part of the optimization objective. It is unclear if the framework naturally discovers diverse functional mechanisms or just structurally different variants that map to the same local optima.

**Suggestion:** Provide an analysis of the functional diversity of the discovered hypotheses, rather than just structural embedding distances.

5. The prompts used for the hot and cold pools in molecular discovery explicitly instruct the LLM to focus on different goals (exploration vs. higher score). This prompt engineering might confound the measured benefits of the thermodynamic swapping mechanism.

**Suggestion:** Run an ablation where both pools use the exact same prompt to isolate the effect of the PT swap.

6. The paper mentions invalid proposals are rejected and not counted towards the oracle budget. If the hot pool generates significantly more invalid proposals, the wall-clock time could be much higher despite an equal valid budget.

**Suggestion:** Report the invalid proposal rates for each method and pool to provide a complete picture of efficiency.

7. The visualization in Figure 11 shows PCA projections, but PCA can sometimes distort local neighborhood structures compared to t-SNE or UMAP. This might affect the visual interpretation of population overlap.

**Suggestion:** Provide t-SNE or UMAP visualizations for the algorithm discovery task in the appendix for comparison.

---

> ### Author Rebuttal · Authors · 2026-03-31
>
> We are extremely appreciative of the reviewer’s feedback and suggestions to improve our work. We are glad that the reviewer finds our method "theoretically sound" with good generalizability. We provide our new results here: https://drive.google.com/file/d/13HP1G8MK6tgw-2QTmg2STuV3DJICNWJD/view?usp=sharing
>
> **All figure and table references below correspond to those in the new document unless otherwise specified.**
>
> > Q1: Sensitivity to ξ initialization and target swap rate?
>
> **Figure 1** shows diagnostic curves of $\xi$ and acceptance rate over iterations for molecular discovery. The adaptive controller adjusts $\xi$ to maintain an acceptance rate within 30–50% regardless of initialization, consistent with PT conventions.
>
> **Tables 5–6** present sensitivity studies on equation discovery (10 Physics problems, DeepSeek-V3.2). We vary $\beta_1$ ∈ {0.5, 0.8, 1.0, 1.5}, $\beta_2$ ∈ {0.01–0.5}, swap period ∈ {2, 5, 10, 20}, and k ∈ {1, 3, 5, 10}. Performance is stable across a wide range (e.g., $\beta_1$ ∈ {0.8, 1.0} all achieve Best $Acc_{₀.₁}$ ≥ 0.4), degrading only at extremes ($\beta_2$ = 0.5 or $\beta_1$ = 1.5). More frequent swapping (period 2–5) and k ≥ 5 work best, with graceful degradation otherwise.
>
> > Q2: Why only two temperature pools?
>
>  Our choice of using only two temperature pools was mainly empirical. Since we keep the total computation budget fixed across methods for a fair comparison, using more temperature levels would split the budget more thinly, giving each pool fewer updates and potentially hindering convergence.
>
>  In **Table 2**, we also test 3 temperatures ($\beta$ ∈ {0.8, 0.2, 0.01}) on equation discovery: diversity improves slightly (0.316 vs. 0.308) with comparable quality (Best $Acc_{₀.₁}$: 0.421 vs. 0.416). **This confirms the framework generalizes to more temperatures**, while two pools offer the simplest setup with an overall good cost-benefit tradeoff.
>
> > Q3: Are per-generation budgets strictly identical across methods?
>
> Yes. **Table 7** details per-generation parent/offspring counts, pool sizes, and oracle budgets for all methods across all domains. All methods share the same total budget; multi-pool methods split it evenly. The MH swap consumes no additional oracle evaluations.
>
> > Q4: Functional diversity vs. merely structural?
>
> Evidence from all three domains:
> - **Molecular discovery** (Figure 2b, main paper): candidates are filtered by two *unseen* objectives (QED > 0.5, SA < 5.5) never optimized during search. EvoDiverse passes these filters while maintaining high scores, showing functional distinctness along drug-likeness and synthesizability.
> - **Algorithm discovery** (Figure 4, main paper): EvoDiverse discovers qualitatively different strategies—hexagonal grid, border-first greedy, simulated annealing, iterative refinement—representing functionally distinct approaches, not syntactic variations.
> - **Equation discovery**: CodeBERT embeddings capture semantic structure. EvoDiverse achieves higher diversity *and* better Top-10 accuracy simultaneously—if diversity were syntactic noise, it would not improve quality.
>
> We acknowledge that fully rigorous behavioral equivalence testing is valuable future work.
>
> > Q5: Prompt effect vs. MH swap mechanism?
>
> **Table 3** (same-prompt ablation): both pools use the identical cold-pool prompt, differing only in selection temperature $\beta$. Results: on JNK3, same-prompt EvoDiverse achieves Top-10 AUC **0.65** (vs. 0.63 with different prompts, 0.58 for MOLLEO); on GSK3$\beta$, **0.83** (vs. 0.76 and 0.70). **The gain is driven by the PT mechanism, not prompt engineering.**
>
> > Q6: Invalid proposal rates and wall-clock time?
>
> **Tables 3–4** report full computational statistics. Invalid rates are uniformly low (0.4–1.2%) across all methods. The hot pool has slightly higher rates (e.g., 0.8% vs. 0.4% on JNK3), but this is overall negligible. Wall-clock time is comparable—EvoDiverse (3.82h on JNK3) is actually slightly faster than MOLLEO (4.06h) as the two pools are run in parallel. The MH swap involves only score comparisons with no extra LLM calls.
>
> > Suggestion: Dedicated Limitations section
>
> Agreed. We will add a Limitations section discussing: (1) computational overhead of multiple pools (shown to be minimal), (2) prompt sensitivity (shown to be secondary via ablation), and (3) the need for real-world experimental validation.
>
> > Suggestion: t-SNE visualization for algorithm discovery
>
> **Figure 2** provides t-SNE of final populations on circle packing using TF-IDF embeddings. The patterns match PCA (Appendix Figure 11): EvoDiverse shows differentiated but overlapping pools; Island is homogenized; Ensemble is disconnected.
>
> We hope these new experiments address the reviewer's concerns. We would be grateful for a reassessment if the reviewer finds our results strengthen the paper.

---

> > ### Author Rebuttal · Reviewer_ocBA · 2026-04-04
> >
> > I acknowledge the authors' thorough rebuttal. The newly provided ablation studies on ξ sensitivity (Tables 5–6), the same-prompt ablation (Table 3), the per-generation budget breakdown (Table 7), and the invalid proposal rate analysis effectively address my main technical concerns (Q1, Q3, Q5, Q6). The three-temperature experiment (Table 2) provides a reasonable empirical justification for the two-pool design (Q2), though a brief theoretical discussion in the revised paper would further strengthen the narrative.
> >
> > Regarding functional diversity (Q4), the evidence from unseen-objective filtering in molecular discovery and the qualitatively distinct strategies in algorithm discovery is encouraging, though not fully conclusive. I appreciate the authors' candor in noting this as future work.
> >
> > Overall, the rebuttal has substantively strengthened the paper. I am adjusting my score upward accordingly.

---

> > > ### Author Response · Authors · 2026-04-08
> > >
> > > Thank you for the thoughtful engagement and for raising the score! We will include a discussion on the rationale behind the temperature choice and number of pools, along with connections to the broader parallel tempering literature. We also agree that functional diversity deserves more validation. We will add this to future work, including wet lab validation for molecular discovery.

---

### Official Review · Reviewer_HFZg · 2026-03-10

**Soundness:** 3
**Presentation:** 3
**Significance:** 3
**Originality:** 3
**Overall Recommendation:** 5
**Confidence:** 4

**Summary:**

The paper's major contribution is EvoDiverse, an LLM‑guided evolutionary search framework inspired by parallel tempering, a classical sampling technique. EvoDiverse maintains multiple evolutionary temperatures , a high‑temperature population that explores broadly with low selection pressure, and a low‑temperature population that refines promising hypotheses more aggressively. The method includes a principled swap mechanism via a Metropolis–Hastings–style rule that transfers information between these populations without disrupting convergence. This design ensures that exploration and exploitation reinforce each other instead of causing premature convergence or diversity collapse.

The paper evaluates EvoDiverse on three scientific hypothesis search problems: molecular discovery, equation discovery, and algorithm discovery. In all settings, EvoDiverse consistently produces candidate hypothesis sets that are both higher‑quality and more diverse than those generated by standard LLM‑based evolutionary baselines such as MOLLEO, high‑temperature tempering, and ensemble/island evolutionary setups.

**Compliance With Llm Reviewing Policy:**

Affirmed.

**Final Justification:**

The authors have thoroughly addressed all concerns in their rebuttal responses, offering deeper and more detailed clarification of their proposed research. These has contributed to the overall recommendation in the final evaluation score from Weak Accept to Accept.

**Key Questions For Authors:**

1. The paper would benefit from additional ablation studies that isolate the contribution of key components. Can the authors report the results from ablation study removing the Metropolis–Hastings swap to clarify how much diversity and convergence stability arise from this component?
2. All experiments are conducted using general‑purpose LLMs. Can the authors provide a rationale  to determine whether EvoDiverse’s gains are intrinsic to the framework or partly dependent on the capabilities of the underlying LLMs?
3. Can the authors kindly clarify the computational cost of diversity, for e.g., how much extra computation is required per additional unit of diversity gained?

**Limitations:**

Yes.

**Strengths And Weaknesses:**

Strengths:
1. The paper’s theoretical framing from a Boltzmann‑like distribution aligns with established ideas in sampling theory and parallel tempering and the dynamic adaptation of ξ(n) is a pragmatic solution for stabilizing swap acceptance rates under a non‑stationary target distribution.
2. The paper clearly highlights the problem of diversity collapse in LLM‑guided evolutionary search.
3. EvoDiverse is evaluated across three structurally distinct scientific problems across three different domains to support generality.

Weaknesses:
1. All evaluations rely on general‑purpose LLMs (e.g., DeepSeek‑V3.2 and GPT‑5) rather than models fine‑tuned for the specific scientific domains.
2. The paper contains a noticeable number of typographical and formatting errors.
3. EvoDiverse explicitly maintains multiple temperature‑level populations and performs frequent Metropolis–Hastings–style swaps, which increases computational overhead relative to standard LLM‑guided evolutionary algorithms. While the method clearly improves hypothesis diversity, the paper does not thoroughly discuss the trade‑off between achieving this diversity and the additional runtime.

---

> ### Author Rebuttal · Authors · 2026-03-31
>
> We thank the reviewer for their positive feedback! We are thrilled by the reviewer's positive assessment of the paper’s main idea and for recognizing the value of the sampling perspective and the MH-style exchange mechanism.
> We provide our new results here: https://drive.google.com/file/d/1-o0RK5GU_-lEdZIfU55OehMrXca8QlFj/view?usp=sharing
>
> **All figure and table references below correspond to those in the new document.**
>
> > W1 & Q2: General-purpose LLMs / Whether gains are intrinsic to the framework
>
> Our choice follows the established protocol in LLM-augmented evolutionary search (MOLLEO, LLM-SR, FunSearch), which all use general-purpose models. EvoDiverse is a **framework-level** contribution agnostic to the base model. Three lines of evidence:
>
> 1. **Cross-backbone consistency (Appendix Table 6):** EvoDiverse improves over baselines with both DeepSeek-V3.2 and GPT-5 in equation discovery.
> 2. **Non-LLM algorithm (Appendix Table 4):** EvoDiverse also improves GraphGA, a heuristic-based EA using no LLM (JNK3 AUC: 0.35→0.43; GSK3β: 0.59→0.67).
> 3. **Ablation results below** confirm the gains come from the PT architecture and MH swap — model-agnostic components.
>
> Domain-specific fine-tuning is orthogonal and can be combined with EvoDiverse for additional gains.
>
>
> > W2: Typo errors
>
> We will fix all issues in the next version.
>
> > W3 & Q3: Computational overhead / Cost of diversity
>
> All methods share the same fixed oracle budget. For multi-pool methods, the budget is split across pools. Therefore, all the comparisons are fair under the same oracle budget. The only additional overhead for our approach is the MH swap — simple arithmetic (log-acceptance ratios), negligible vs. LLM inference.
>
> Tables 1 and 2 report LLM calls, tokens, wall-clock time, and invalid rates for all methods on JNK3 and GSK3$\beta$. EvoDiverse uses comparable LLM calls/tokens with similar or shorter wall-clock time, while achieving the best Top-10 AUC and the lowest invalid rate. MOLLEO uses fewer calls on GSK3β because it early-stops at a suboptimal solution. **The diversity gain is essentially free** — it comes from better allocation of the same budget, not additional computation.
>
> > Q1: Ablation isolating MH swap contribution
>
> We compare three configurations across all three domains: **Ensemble** (no swap), **Random Swap** (same two-pool architecture, all swaps accepted unconditionally without MH gate), and **EvoDiverse** (full MH acceptance gate). Results are in Tables 3, 4, and 5.
>
> **Key findings across all three domains:**
>
> 1. **Swapping helps:** Both Random Swap and EvoDiverse outperform Ensemble in optimization quality, confirming inter-pool communication is beneficial.
>
> 2. **MH gate improves stability and quality:** EvoDiverse achieves better or comparable quality with notably lower variance than Random Swap (e.g., JNK3 Top-10 AUC std: 0.05 vs. 0.08). The MH criterion selectively transfers candidates consistent with both temperatures, preventing disruption of the cold pool's convergence.
>
> 3. **MH gate preserves diversity:** In circle packing, Random Swap's diversity (0.576) drops *below* even single-pool EA (0.596) — unconditional injection of cold-pool candidates homogenizes the hot pool. The MH gate prevents this, achieving the highest diversity (0.714). In equation discovery (Physics), EvoDiverse's diversity (0.308) also exceeds Random Swap (0.295).
>
> These results confirm that the MH acceptance gate is a critical component: it enables productive information exchange while preventing the disruption that naive migration causes.
>
> We hope our new experiments and analyses address the reviewer's concerns. We would be grateful if the reviewer would consider updating their assessment in light of these results. We remain more than happy to address any additional questions or concerns that may arise. Thank you once again for your time and thoughtful feedback.

---

> > ### Author Rebuttal · Reviewer_HFZg · 2026-04-01
> >
> > Thanks to the authors for comprehensively addressing all the concerns raised. In the light of new experiments and the insights provided through those, I am fully satisfied with the responses and am happy to raise my score to 5 (Accept).

---

> > > ### Author Response · Authors · 2026-04-02
> > >
> > > Thank you very much for the positive feedback and for raising the score! We truly appreciate the time and effort you spent reviewing our work.
> > >
> > > Just a gentle note: whenever convenient, it would be great if you could also update the overall recommendation score in the original review. Thanks again!

---

### Official Review · Reviewer_FaYx · 2026-03-13

**Soundness:** 3
**Presentation:** 3
**Significance:** 3
**Originality:** 3
**Overall Recommendation:** 5
**Confidence:** 3

**Summary:**

This work proposes EvoDiverse an evolutionary framework inspired by the Parallel Tempering Algorithm. The proposed method uses a cold and hot pool of candidates with a swapping mechanism between the pool to improve exploration without disrupting convergence. The method is evaluated on three domains: molecular discovery, equation discovery and algorithm discovery, and is shown to improve the quality and diversity of the proposed hypotheses.

**Compliance With Llm Reviewing Policy:**

Affirmed.

**Final Justification:**

The authors have addressed my main concerns in the rebuttal, which included non-diversity-aware evaluations and sensitivity analyses for the temperatures and swap period, as well as the evolution of the powering factor and swap acceptance rate. Given the initial submission and these additions, I am increasing my score from 4 to 5 (Accept).

**Key Questions For Authors:**

1. Could the authors include a table with the non-diversity-aware scores?
2. How many invalid proposals were generated by the different methods? Was the proposed approach more prone to generating an invalid proposal? "All optimization is restricted to $\mathcal{M}_{valid}$; any invalid proposal is rejected and is not counted as an oracle evaluation." (Appendix C.1.1, loc 696-697)
3. Could the authors provide further information on the adaptation step of the $\xi$ parameter?
4. Why were  "4.1. Molecular Discovery" and "4.3. Algorithm Discovery" evaluated with a single LLM backbone, while "4.2. Equation Discovery" was evaluated using both the DeepSeek V3.2 and GPT 5 models?
5. Could the authors provide a figure displaying the evolution of the swap acceptance rate and powering factor for one of the experimental configurations?

**Limitations:**

The method introduces multiple hyperparameters, including $\xi, \tau, \epsilon, \beta_1, \beta_2$  to name a few. How much does the method depend on these?

**Strengths And Weaknesses:**

**Soundness**

- The proposed method is evaluated over multiple domains (molecular discovery, equation discovery, and algorithm discovery) and shows improved performance when compared to the baselines. The different methods are compared with a similar budget.
- Nonetheless, two of the domains use a single LLM backbone for evaluation. While the results in Table 5 show consistent improvement across backbones, the evaluation on a single backbone for the other two tasks raises questions. Why was the GPT 5 backbone not evaluated for the other two domains?
- An overview of the evolution of the swap rate and powering factor for one of the tasks would help better understand the contribution of these components.
- An analysis of the impact of different hyperparameters, such as pool temperatures and swap rates, would help better understand which part of the proposal delivers the greatest benefits.
- For the molecular discovery task, results are reported using discovery-aware metrics. While the authors argue that diversity is important in this setting, it would be interesting to also report non-diversity-aware metrics.
- The appendix mentions that invalid proposals are not counted toward oracle evaluation (see questions in the Key Questions for Authors section). Why is that the case? Was this a recurring phenomenon?

**Presentation**

- The submission is well structured, and the narrative is relatively easy to follow. Nonetheless, a concrete example of the notion of diversity could be included earlier in the paper, as it would provide the reader with a clearer picture of what the concept means across different domains.
- While the appendix contains details of the experimental setup, the algorithm used to adapt the powering factor could be better described.

**Significance**

- The paper proposes a principled approach to balancing exploration and exploitation in hypothesis search. The method is evaluated across three main domains, showing versatility and, as such, could be relevant to a broader community.
- The related work section could be improved by more clearly positioning the proposed approach relative to contemporary methods, as it currently reads more like a summary of the field than a clear articulation of the current work within its ecosystem.

**Originality**

- The motivation for the method is well-articulated, and the benefits of the two-pool system and the swapping mechanism are demonstrated through empirical validation. The adaptation of parallel tempering to LLM-based hypothesis search is a principled contribution; nonetheless, the related work section could better explain what differentiates the proposed approach from existing population-based search methods.

---

> ### Author Rebuttal · Authors · 2026-03-31
>
> We thank the reviewer for their constructive feedback and for recognizing our method as principled and well-motivated. We provide our new results here: https://drive.google.com/file/d/1IKQGMGes20LSO7HUHMs3XyJoAiPJ21H_/view?usp=sharing.
>
> **All figure and table references below correspond to those in the new document.**
>
> > Q1: Non-diversity-aware scores?
>
> Table 1 reports standard Top-10/Top-100 avg scores (ranked purely by score, no diversity filtering):
>
> JNK3: MOLLEO 0.79/0.62, Ensemble 0.75/0.66, Tempering 0.63/0.44, **EvoDiverse 0.82/0.69**
> GSK3β: MOLLEO 0.87/0.71, Ensemble 0.88/0.79, Tempering 0.75/0.54, **EvoDiverse 0.92/0.81**
>
> EvoDiverse achieves the best scores under both diversity-aware (Table 1) and standard metrics, confirming that diversity gains do not come at the cost of quality.
>
> > Q2: Invalid proposal rates?
>
> Tables 2 and 3 report invalid rates alongside computational costs. Key results:
>
> - Molecular (JNK3): MOLLEO 0.7%, Ensemble 1.0%, Tempering 1.2%, **EvoDiverse 0.6%**
> - Molecular (GSK3β): MOLLEO 0.6%, Ensemble 0.6%, Tempering 0.9%, **EvoDiverse 0.6%**
>
> EvoDiverse is *not* more prone to invalid proposals — its rate is consistently among the lowest. This might be because the MH swap selectively transfers higher-quality candidates, preventing borderline-invalid structures from entering the refinement pipeline. Wall-clock times are also comparable across methods under the same oracle budget (e.g., JNK3: MOLLEO 4.06h, EvoDiverse 3.82h), confirming negligible computational overhead.
>
> > Q3: Further information on $\xi$ adaptation?
>
> The adaptive mechanism (Algorithm 2, lines 14–19) maintains a sliding window of recent swap decisions. If the empirical acceptance rate exceeds [$\tau$−$\xi$/2, $\tau$+$\xi$/2], $\xi$ is increased by a factor of 1.1 (more selective swaps); if below, decreased by 0.9 (more permissive). Figure 1 shows diagnostic curves on molecular discovery: $\xi$ increases monotonically (consistent with sharpening distributions requiring larger scaling), while the acceptance rate quickly stabilizes to ~30–40% and remains stable throughout. Convergence to similar regimes across targets confirms stability. We will improve this description in the revised manuscript.
>
> > Q4: Why a single backbone for molecular/algorithm discovery?
>
> This was due to API cost constraints. We prioritized depth of analysis (diversity metrics, QED/SA filtering, t-SNE, per-pool breakdowns) over backbone breadth within each domain. Importantly, the cross-backbone consistency observed in equation discovery (Table 5: EvoDiverse improves with both DeepSeek-V3.2 and GPT-5) provides evidence that gains are attributable to the framework rather than a particular model. All methods within each task use the same backbone, ensuring fair comparison. We plan to add GPT-5 results for the remaining domains in the revised version.
>
> > Q5: Evolution of swap acceptance rate and powering factor?
>
> See Figure 1 (referenced in Q3 above), which plots both $\xi$ and acceptance rate over iterations for JNK3 and GSK3β. Both behave as theoretically expected and confirm the adaptive mechanism's stability.
>
> > Q6: Hyperparameter sensitivity
>
> Tables 4 and 5 provide a comprehensive sensitivity study on equation discovery (10 Physics problems, DeepSeek-V3.2):
>
> **Temperatures:** β₁∈{0.8,1.0} and β₂≤0.1 all perform well. Degradation occurs only at extremes (β₁=1.5 collapses diversity; β₂=0.5 eliminates the temperature gap).
> **Swap period:** 2–5 works well; longer periods (10–20) reduce diversity due to infrequent exchange.
> **Batch size k:** Robust for k≥3, with diminishing returns beyond k=5.
> **Target rate $\tau$:** Follows PT convention (30–50%) and is automatically maintained by adaptive $\xi$, requiring no task-specific tuning.
>
> The defaults ($\beta_1$=0.8, $\beta_2$=0.01, period=5, k=5) work across all three domains without task-specific adjustment, showing graceful degradation rather than catastrophic failure at non-default settings.
>
> > Presentation suggestions
>
> We would like to further thank the reviewer for this valuable suggestion. We will incorporate all three in the revised manuscript: (1) an early concrete example of diversity across domains, (2) expanded $\xi$ adaptation description, and (3) polishing the related work by clearer positioning our approach relative to existing population-based methods and putting the current summary of the field in the appendix.
>
>
> We hope that our new experiments were sufficient in clarifying all the great questions asked by the reviewer. We sincerely thank the reviewer for their time and consideration. Should you find our response satisfactory, we kindly invite you to consider further raising your rating. We would be happy to engage in further discussions if the reviewer has any additional questions or concerns.

---

> > ### Author Rebuttal · Reviewer_FaYx · 2026-04-03
> >
> > I thank the authors for addressing my concerns. Given the clarifications and supporting tables, especially Tables 4, 5, and Figure 1, I am raising my score from weak accept to accept.

---

> > > ### Author Response · Authors · 2026-04-08
> > >
> > > We sincerely thank the reviewer for the thoughtful engagement throughout the discussion period and for raising the score to Accept. We are delighted that the new sensitivity analyses, non-diversity-aware evaluations, and diagnostic curves addressed your concerns. Your constructive feedback has meaningfully strengthened our paper, and we are very grateful for the time and care you devoted to our work. Thank you!

---

### Official Review · Reviewer_MRD1 · 2026-03-16

**Soundness:** 3
**Presentation:** 3
**Significance:** 3
**Originality:** 3
**Overall Recommendation:** 4
**Confidence:** 4

**Summary:**

The paper proposes EvoDiverse, a parallel-tempered evolutionary framework for large language model (LLM)-guided scientific hypothesis search that explicitly targets both quality and diversity under a fixed validation budget. The key idea is to view evolutionary search as approximate sampling from a Boltzmann-like distribution with an evolving power factor and to run two populations at different “temperatures,” periodically exchanging candidates via a Metropolis–Hastings-style swap that aims to preserve each population’s characteristic exploration/exploitation balance. Across molecular discovery, equation discovery, and algorithm discovery, EvoDiverse is reported to improve both solution quality and diversity compared to single-population and ensemble baselines.

**Compliance With Llm Reviewing Policy:**

Affirmed.

**Key Questions For Authors:**

1. How sensitive is EvoDiverse to the choice of β1, β2, swap period, swap batch size k, and the target acceptance rate τ? Could you include a sensitivity study or guidance for setting these across tasks?
2. Can you ablate the MH-style swap against simpler migration rules (e.g., periodic injection without acceptance) in all three domains to quantify the added value of the acceptance gate?
3. You mention potentially using different prompts at different temperatures. Do the gains hold when prompts are identical across temperatures, isolating the effect of PT-style exchange?
4. What is the exact number of runs per experiment (and seeds), and are the observed gains statistically significant (e.g., paired tests) across runs?
5. In molecular discovery, beyond t-SNE and QED/SA, can you report scaffold-level novelty/diversity metrics and any secondary validation (e.g., docking, physics-based scoring) to support the “robust under more expensive validations” claim?
6. How do total LLM calls and token costs compare across methods under the fixed oracle budget? Is compute/time-to-result comparable for EvoDiverse vs ensemble/single-pool baselines?
7. Could you compare against a quality-diversity baseline (e.g., MAP-Elites or novelty-search variants) to contextualize EvoDiverse’s diversity–quality trade-offs?
8. Can you share diagnostic curves (e.g., acceptance rates and ξ over iterations) to illustrate stability of the adaptive swap-rate controller and whether it converges to similar regimes across runs?
9. In equation discovery, how do you ensure that diversity is not inflated by syntactic variations of semantically equivalent programs? Do you canonicalize expressions before measuring diversity?

**Limitations:**

yes

**Strengths And Weaknesses:**

The paper makes a technically novel contribution by framing LLM-augmented evolutionary search as approximate sampling and introducing a parallel-tempering-inspired exchange mechanism with adaptive acceptance to mitigate diversity collapse—a simple, modular approach applicable across molecules, symbolic programs, and algorithms. The experimental evaluation is rigorous, covering three discovery domains with appropriate diversity-aware metrics, multiple baselines, and two LLM backbones, showing faster convergence and stronger candidate sets under fixed evaluation budgets while maintaining competitive drug-likeness and synthetic accessibility scores. The presentation is clear, and the work addresses an important practical challenge—diversity collapse in LLM-driven hypothesis search—offering a drop-in upgrade for existing LLM-based evolutionary algorithms in scientific workflows.

---

> ### Author Rebuttal · Authors · 2026-03-31
>
> We thank the reviewer for the thorough feedback and recognition of our novelty, rigorous evaluation, and practical contribution. We updated the new results here: https://drive.google.com/file/d/1L5oletjPkmouy4Pox_L2uhug3cPTf2JR/view?usp=sharing. **Figure/table references below correspond to those in the new document.**
>
> >Q1: Sensitivity to parameters.
>
> Tables 1–2 show a sensitivity study on equation discovery (10 Physics problems, DeepSeek-V3.2). Performance is stable across $\beta_1$ $\in$ {0.8, 1.0} and $\beta_2$ $\leq$ 0.1, degrading only at extremes ($\beta_1$=1.5 collapses diversity, $\beta_2$=0.5 loses exploration). Swap period 2–5 works well; batch size k is robust for $k \geq 3$. The target acceptance rate $\tau$ follows PT convention (30–50%) and is automatically maintained by our adaptive $\xi$ mechanism (see Q8). In summary, the defaults ($\beta_1$=0.8, $\beta_2$=0.01, period=5, k=5) work across all three domains without task-specific tuning.
>
> >Q2: Ablate MH-style swap.
>
> Tables 3–5 compare EvoDiverse against "Random Swap" (i.e, no MH) across all three domains. Key findings: (1) On algorithm discovery, EvoDiverse achieves the best sum 2.5461 vs. Random Swap's 2.4507, with much higher diversity (0.714 vs. 0.576)—without the MH gate, strong cold-pool solutions dominate the hot pool. (2) On equation discovery (Physics), Best $Acc_{0.1}$ improves from 0.346 to 0.416 (+20%). (3) On molecular discovery, MH swap substantially reduces variance (std 0.04 vs. 0.09 on JNK3). **The MH is critical for preventing disruptive exchanges while enabling productive information flow.**
>
> >Q3: Same-prompt ablation.
>
> Table 6 shows that with identical prompts in both pools (only temperature differs), EvoDiverse still achieves strong gains over MOLLEO: +12% Top-10 AUC on JNK3, +19% on GSK3$\beta$. **This confirms gains are intrinsic to the PT mechanism, not prompt engineering. Prompt specialization is an optional, complementary enhancement.**
>
> >Q4: Number of runs and statistical significance.
>
> Molecular and algorithm discovery: 3 seeds (0,1,2). Equation discovery: averaged over 129 problems × 2 backbones. New Tables 7–8 report paired one-sided t-tests. On equation discovery, EvoDiverse is significant over all baselines (p < 0.001). On molecular discovery, significant over MOLLEO (p=0.011, JNK3) and Tempering (p=5.4e-6, GSK3$\beta$).  One exception is the Ensemble. However, we note that this result may be inconclusive due to the small sample size. Moreover, Table 9 shows that Ensemble achieves a lower QED on JNK3 (0.12 vs. 0.21), suggesting reward hacking for Ensemble.
>
> >Q5: Scaffold-level metrics and secondary validation.
>
> Table 9 reports Murcko scaffolds, scaffold diversity, QED, and SA. EvoDiverse maintains high scaffold counts (9.67–10) with superior QED, while Ensemble shows low QED indicating chemically unrealistic molecules. Table 10 confirms EvoDiverse's advantage holds under three different diversity-constrained selection criteria. Table 11 reports UniDock docking: EvoDiverse achieves the best JNK3 docking score (−7.83 ± 0.08 kcal/mol vs. MOLLEO −7.53 ± 0.80) with the lowest variance, confirming robustness under expensive physics-based validation.
>
> >Q6: LLM calls and token costs.
>
> Tables 12–13 report computational costs. Under the same 10,000 oracle budget, all methods consume comparable resources (8–10M tokens, ~4h wall-clock on JNK3). EvoDiverse's wall-clock time is comparable to or lower than baselines (3.82h vs. 4.06h) since two pools enable parallelism. Invalid rates are low across all methods (<1.2%), with EvoDiverse showing the lowest (0.4–0.6%). Computational overhead is negligible.
>
> >Q7: Compare against MAP-Elites.
>
> MAP-Elites requires predefined behavior descriptors, which is itself a research challenge for molecules and symbolic equations. EvoDiverse achieves diversity without domain-specific descriptors. Our framework is compatible with QD-integrating MAP-Elites archives into each pool, which is a promising future direction. The Ensemble baseline serves as the closest diversity-preserving comparator in our evaluation.
>
> >Q8: Diagnostic curves.
>
> Figure 1 shows $\xi$ and the acceptance rate over iterations. The acceptance rate quickly stabilizes to ~30–40%, and $\xi$ increases monotonically—consistent with motivation in our paper. Convergence to similar regimes across targets confirms stability.
>
> >Q9: Syntactic vs. semantic diversity in equation discovery.
>
> We mitigate by: (1) extracting only the mathematical function body (stripping docstrings/comments) before computing embeddings; (2) using CodeBERT embeddings because they capture program semantics rather than surface syntax, grouping functionally similar programs even with different syntax. Full symbolic canonicalization is a promising extension we will discuss in the revision.
>
> Should you find our response satisfactory, we kindly invite you to consider further raising your rating. We are happy to address any additional comments the reviewer may have.

---

> > ### Author Rebuttal · Reviewer_MRD1 · 2026-04-03
> >
> > I remain with my weak accept. I still perceive the work as somewhat valuable, but also not a breakthrough.

---

> > > ### Author Response · Authors · 2026-04-08
> > >
> > > We thank the reviewer for the continued engagement and for maintaining the positive assessment.
> > >
> > > We would briefly highlight our core contributions: we formulate LLM-guided evolutionary search as approximate Boltzmann sampling and introduce an adaptive Metropolis-Hastings swap that, unlike classical island EA's heuristic migration, enables principled cross-temperature communication. Our method consistently improves in both quality and diversity across three different scientific domains. For example, in real-world molecular discovery, wet-lab validation involves many unobserved objectives beyond the optimized score. We were motivated by this challenge to leverage diverse candidate sets to hedge against such unobserved factors, as demonstrated by the QED/SA and docking results in our rebuttal.
> > >
> > > We thank the reviewer again for the valuable feedback, which has meaningfully improved our paper.

---

### Decision · Program_Chairs · 2026-04-30

**Decision:**

Accept (regular)

**Comment:**

The paper tackles an important problem, and the empirical results are overall quite promising across multiple domains. My main reservation is about the equation-diversity metric: CodeBERT is trained on GitHub code corpora rather than designed for algebraic equivalence, so it is not obvious that it can reliably treat mathematically equivalent expressions such as (a(b+c)) and (ab+ac) as the same. Still, I see this more as a caveat about how strongly to interpret the diversity claim than a reason to dismiss the paper. The work is well motivated, the method is reasonably simple and general, and the experiments suggest the idea is practically useful. I also think it has a good chance of inspiring follow-up work in this area. For those reasons, I lean towards accepting.